# How enhancers regulate wavelike gene expression patterns

Christine Mau[1], Heike Rudolf[1], Frederic Strobl[2], Benjamin Schmid[3], Timo Regensburger[1], Ralf Palmisano[3], Ernst HK Stelzer[2], Leila Taher[4]*, Ezzat El-Sherif[5]*

[1]Division of Developmental Biology, Department of Biology, Friedrich-Alexander-Universität Erlangen-Nürnberg, Erlangen, Germany; [2]Buchmann Institute for Molecular Life Sciences (BMLS), Goethe Universität Frankfurt Am Main, Frankfurt am Main, Germany; [3]Optical Imaging Centre Erlangen (OICE), Friedrich-Alexander-Universität Erlangen-Nürnberg, Erlangen, Germany; [4]Institute of Biomedical Informatics, Graz University of Technology, Graz, Austria; [5]Department of Biology, University of Texas Rio Grande Valley, Edinburg, United States

*For correspondence:
leila.taher@tugraz.at (LT);
ezzat.elsherif@utrgv.edu (EE-S)

**Competing interest:** The authors declare that no competing interests exist.

## Abstract

A key problem in development is to understand how genes turn on or off at the right place and right time during embryogenesis. Such decisions are made by non-coding sequences called 'enhancers.' Much of our models of how enhancers work rely on the assumption that genes are activated de novo as stable domains across embryonic tissues. Such a view has been strengthened by the intensive landmark studies of the early patterning of the anterior-posterior (AP) axis of the *Drosophila* embryo, where indeed gene expression domains seem to arise more or less stably. However, careful analysis of gene expression patterns in other model systems (including the AP patterning in vertebrates and short-germ insects like the beetle *Tribolium castaneum*) painted a different, very dynamic view of gene regulation, where genes are oftentimes expressed in a wavelike fashion. How such gene expression waves are mediated at the enhancer level is so far unclear. Here, we establish the AP patterning of the short-germ beetle *Tribolium* as a model system to study dynamic and temporal pattern formation at the enhancer level. To that end, we established an enhancer prediction system in *Tribolium* based on time- and tissue-specific ATAC-seq and an enhancer live reporter system based on MS2 tagging. Using this experimental framework, we discovered several *Tribolium* enhancers, and assessed the spatiotemporal activities of some of them in live embryos. We found our data consistent with a model in which the timing of gene expression during embryonic pattern formation is mediated by a balancing act between enhancers that induce rapid changes in gene expression patterns (that we call 'dynamic enhancers') and enhancers that stabilize gene expression patterns (that we call 'static enhancers'). However, more data is needed for a strong support for this or any other alternative models.

## Editor's evaluation

The authors describe a sophisticated method to follow enhancer activity in both live embryos and fixed embryos in Tribolium and present important data about the function of a number of enhancers in early development. They show that some of the enhancers are "dynamic" and others are "static" and use this to provide solid support for the "enhancer-switching" model of gene regulation suggested by some of these authors in the past. Additional work is required to provide conclusive validation of the "enhancer switching" model.

## Introduction

While an embryo is growing, each cell continuously receives inputs from surrounding cells. The cell processes these inputs and decides its fate accordingly. This decision-making process relies on non-coding sequences called 'enhancers' (*Spitz and Furlong, 2012*; *Shlyueva et al., 2014*). Much of our models of how enhancers work during development relies on the assumption that genes are activated de novo across embryonic tissues as stable domains of gene expression (*Small et al., 1992*; *Clyde et al., 2003*; *Hoch et al., 1991*), that then undergo little or no change, either indefinitely or until they do their job whereafter they gradually fade away. Such a view has been strengthened by the intensive landmark studies of the early patterning of the AP axis of the fruit fly *Drosophila melanogaster* embryo, where indeed gene expression domains seem to arise more or less stably (reviewed in *Clark et al., 2019*; *El-Sherif et al., 2012b*; *Jaeger, 2011*; *Diaz-Cuadros et al., 2021*). However, careful analysis of gene expression patterns in other model systems painted a different, very dynamic view of gene regulation (*Diaz-Cuadros et al., 2021*; *Di Talia and Vergassola, 2022*). For example, during the AP patterning of vertebrates, oscillatory waves of gene expression were shown to sweep the posterior end of the embryo before they stabilize into their final positions (*Diaz-Cuadros et al., 2021*; *Palmeirim et al., 1997*; *Oates et al., 2012*; *Lauschke et al., 2013*; *Aulehla et al., 2008*; *Soroldoni et al., 2014*; *Sonnen and Janda, 2021*), demarcating the somites that eventually give rise to various segmented tissues including vertebrae and skeletal muscles. Likewise, Hox gene expression domains propagate along the AP axis of vertebrates, demarcating future axial identities (*Diaz-Cuadros et al., 2021*; *Forlani et al., 2003*; *Gaunt and Strachan, 1994*; *Deschamps and Duboule, 2017*; *Durston et al., 2012*; *Durston et al., 2010*). In both the neural tube and limb bud of vertebrates, gene expression domains arise in a temporal sequence and spread across the tissue, dividing them into different embryonic fates (*Dessaud et al., 2007*; *Balaskas et al., 2012*; *Harfe et al., 2004*; *Saiz-Lopez et al., 2015*; *Roensch et al., 2013*; *Towers and Tickle, 2009*).

Surprisingly, the patterning of the AP axis of insects, the same process that popularized the static view of gene regulation, turned out to be much more dynamic than previously thought. In insects, the AP axis is divided into segments via the striped expression of a group of genes called 'pair-rule' genes, and into domains of different axial fates via the expression of a group of genes called 'gap genes' (*Clark et al., 2019*; *Diaz-Cuadros et al., 2021*). In the flour beetle *Tribolium castaneum*, a short-germ insect thought to retain a more ancestral mode of AP patterning than long-germ insects like *Drosophila*, both pair-rule and gap genes are expressed as dynamic waves that propagate from posterior to anterior (*El-Sherif et al., 2012a*; *Sarrazin et al., 2012*; *El-Sherif et al., 2014*; *Zhu et al., 2017*; *Boos et al., 2018*). Similar dynamics seem to be involved in segmentation in other insects and arthropods (*Brena and Akam, 2013*; *Pueyo et al., 2008*; *Stollewerk et al., 2003*; *Rosenberg et al., 2014*; *Kanayama et al., 2011*; *Chipman and Akam, 2008*; *Pechmann et al., 2009*). Even pair-rule and gap genes in *Drosophila*, classically thought to be expressed stably, were shown more recently to undergo dynamic (albeit limited) posterior-to-anterior shifts (*Jaeger et al., 2004*; *El-Sherif and Levine, 2016*; *Lim et al., 2018*; *Berrocal et al., 2020*; *Keränen et al., 2006*), a phenomenon that has been suggested (arguably) to be an evolutionary vestige of outright gene expression waves of the sort observed in *Tribolium* and other insects (*Diaz-Cuadros et al., 2021*; *Rudolf et al., 2020*; *Verd et al., 2018*; *Clark and Peel, 2018*; *Clark and Desplan, 2017*; *Stahi and Chipman, 2016*; *Auman and Chipman, 2018*; *Xiang et al., 2017*; *Auman et al., 2017*). These observations show that the static view of gene regulation, once popularized by classical studies of AP patterning in *Drosophila*, is inaccurate and that gene regulation is in most cases a dynamic phenomenon. Hence, new models of embryonic pattern formation – and concomitantly, new models of how enhancers work within pattern formation models – are needed.

Some of the authors have recently suggested a model that explains the generation of either periodic or non-periodic wavelike gene expression patterns, termed the 'Speed Regulation' model (*Figure 1A and B*; *Diaz-Cuadros et al., 2021*; *Zhu et al., 2017*; *Boos et al., 2018*; *Rudolf et al., 2020*; *Kuhlmann and El-Sherif, 2018*). In this model, a morphogen gradient (of a molecular factor termed the 'speed regulator') modulates the speed of either a molecular clock or a genetic cascade. This scheme was shown (in silico) to be able to produce periodic waves in the former case (*Figure 1A*), and non-periodic waves in the latter (*Figure 1B*). The model was shown to be involved in generating pair-rule and gap gene expression waves in the early *Tribolium* embryo where a gradient of Caudal/Wnt was suggested to act as a speed regulator (*El-Sherif et al., 2014*; *Zhu et al., 2017*). These results are consistent with

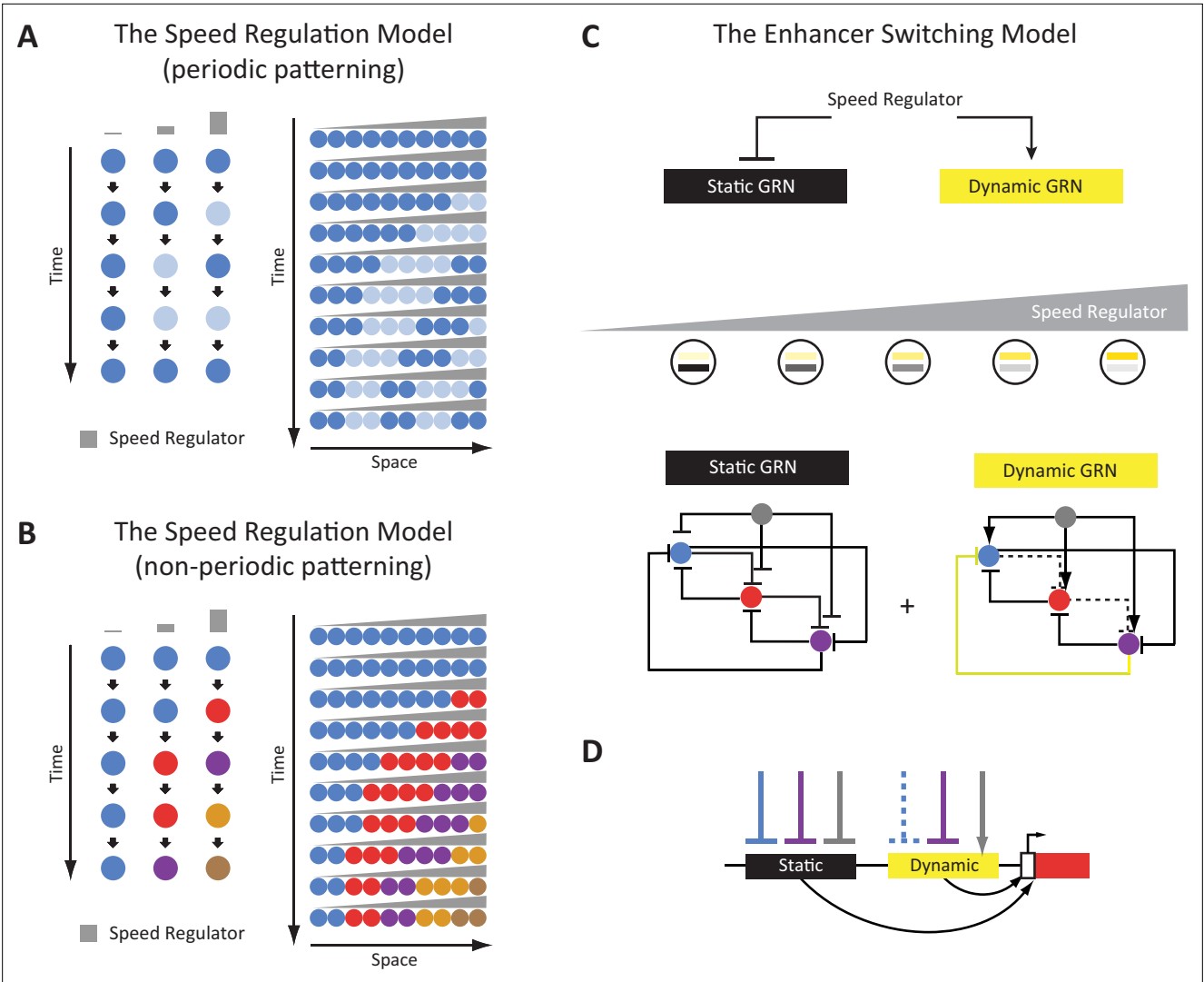

**Figure 1.** The Enhancer Switching Model as a molecular realization of the Speed Regulation Model. (**A**) The Speed Regulation Model for periodic patterning. Left: Cells can oscillate between two states mediated by a molecular clock: high (shown in dark blue), and low (shown in light blue). The concentration of the speed regulator (shown in gray) modulates the speed of the molecular clock (i.e. its frequency). Right: A gradient of the speed regulator across a tissue (represented by a row of cells) induces a periodic wave that propagates from the high to the low end of the gradient. (**B**) The Speed Regulation Model for non-periodic patterning. Same as for the periodic case (**A**), except the molecular clock is replaced with a genetic cascade that mediates the sequential activation of cellular states (represented by different colors). (**C**) The Enhancer Switching Model, a molecular realization of the Speed Regulation model, is composed of two gene regulatory networks (GRNs): one dynamic and one static. The dynamic GRN can be either a clock (to mediate periodic patterning) or a genetic cascade (to mediate non-periodic patterning). The static GRN is a multi-stable gene circuit that mediates the stabilization of gene expression patterns. The speed regulator activates the dynamic GRN but represses the static GRN, and so a gradient of the speed regulator (shown in gray) mediates a gradual switch from the dynamic to the static GRN along the gradient. Shown are example realizations of dynamic and static GRNs, where the dynamic GRN represents either a molecular clock or a genetic cascade, depending on the absence or presence of the repressive interaction shown in yellow, respectively. (**D**) Separate dynamic and static enhancers encode the wiring of each gene (shown here only for the red gene) within the dynamic and static GRNs, respectively.

recent findings in vertebrate somitogenesis (*Falk et al., 2022*; *Diaz-Cuadros et al., 2020*; *Shih et al., 2015*). Furthermore, a molecular model, termed the 'Enhancer Switching' model (*Figure 1C and D*), has been suggested as a mechanism for how a morphogen gradient could fine-tune the speed of a clock or a genetic cascade, serving as a molecular realization of the Speed Regulation model (*Diaz-Cuadros et al., 2021*; *Zhu et al., 2017*; *Kuhlmann and El-Sherif, 2018*). The Enhancer Switching model posits that each patterning gene is simultaneously wired into two gene regulatory networks (GRNs) (*Figure 1C*): (*i*) a dynamic GRN that drives periodic or sequential gene activities, and (*ii*) a static GRN that stabilizes gene expression patterns. The concentration of the speed regulator (shown

in gray in *Figure 1C*) activates the dynamic GRN while repressing the static GRN, and hence sets the balance between the contribution of each GRN to the overall dynamics and, consequently, the speed of gene regulation (*Diaz-Cuadros et al., 2021*; *Zhu et al., 2017*; *Kuhlmann and El-Sherif, 2018*). At high concentrations of the speed regulator, the dynamic GRN is more dominant than the static one, and hence fast oscillations or sequential gene activities are mediated. On the other hand, at low concentrations of the speed regulator, the static GRN is more dominant, and hence slow oscillations or sequential gene activities are mediated. As mentioned, the model posits that each gene is wired into two different GRNs, a requirement that was suggested to be realized using two enhancers per patterning gene: (*i*) a dynamic enhancer that encodes the wiring of the gene within the dynamic GRN, and (*ii*) a static enhancer that encodes the wiring of the gene within the static GRN (*Figure 1D*). The model is partially supported (or rather inspired) by observations in the early *Drosophila* embryo, where the gap gene *Krüppel* (*Kr*) was shown to be regulated by two enhancers whose activities resemble those predicted by the Enhancer Switching model (*Zhu et al., 2017*; *El-Sherif and Levine, 2016*). Similar observations were made for the *Drosophila* gap gene *giant* (*gt*) (*Hoermann et al., 2016*). Furthermore, in vertebrates, it has been suggested that some enhancers or genetic programs mediate the initiation of segmentation clock waves posteriorly, and others mediate their anterior expression domains (*Oginuma et al., 2010*; *Shifley et al., 2008*; *Stauber et al., 2009*).

Despite its potential inaccuracies, the Enhancer Switching model exemplifies the type of alternative frameworks we need to explore in order to elucidate the mechanisms driving the generation of gene expression waves during development. Consequently, an appropriate model system is required, allowing us to test not only the Enhancer Switching model but also any other prospective model that provides a satisfactory explanation for the generation of gene expression waves at the enhancer level. Most enhancers regulating AP patterning have been discovered and characterized in *Drosophila*, and so early patterning of the *Drosophila* embryo might seem like a good model system to study enhancer regulation of dynamic gene expression patterns (*Schroeder et al., 2011*; *Schroeder et al., 2004*). However, gap and pair-rule gene expression domains appear more-or-less de novo in the early *Drosophila* embryo and only undergo 'partial' propagation (usually called 'shifts') form posterior to anterior. This is in contrast to the 'full' gene expression waves of the sort observed during the AP patterning of vertebrates or short-germ insects like *Tribolium*. Hence, we sought to adopt a model system where 'full' gene expression waves are observed. We thought that the AP patterning of *Tribolium* serves our purpose well, and more generally, is an excellent model system to study enhancer regulation of dynamic gene expression patterns. First, *Tribolium* exhibits robust systemic RNAi, which greatly eases the generation of RNAi knockdowns using parental RNAi (*Bucher et al., 2002*; *Miller et al., 2012*; *Tomoyasu et al., 2008*; *Dönitz et al., 2015*). Second, AP patterning takes place in the early *Tribolium* embryo, which eases the interpretation of RNAi knockdowns generated using parental RNAi, without the need for time-specific or tissue-specific genetic perturbations (*Zhu et al., 2017*; *Boos et al., 2018*; *Choe and Brown, 2009*; *Bolognesi et al., 2008a*; *Choe et al., 2006*; *Kotkamp et al., 2010*; *Schoppmeier et al., 2009*; *Cerny et al., 2005*; *Bucher and Klingler, 2004*; *Schmitt-Engel et al., 2012*; *Marques-Souza et al., 2008*; *Savard et al., 2006*; *Marques-Souza, 2007*; *Jeon et al., 2020*). Third, a wide array of genetic and genomic techniques has been developed for *Tribolium* (*Klingler and Bucher, 2022*).

Thus, in this work, we sought to establish the patterning of the early *Tribolium* embryo as a model system for studying enhancer regulation of dynamic and wavelike gene expression patterns. To that end, we set out to (*i*) discover enhancer regions that regulate early patterning genes in *Tribolium*, and (*ii*) characterize the spatiotemporal activity dynamics of these enhancers.

Several strategies can be used to predict enhancer regions, each with their own advantages and disadvantages. Assaying open chromatin is a popular method. In particular, 'Assay for Transposase-Accessible Chromatin with high-throughput sequencing' (ATAC-seq) (*Buenrostro et al., 2013*) is fast and sensitive, and requires very little embryonic tissue (often one embryo, or even a tissue dissected from one embryo) compared to other open chromatin assays. Nevertheless, not all open chromatin regions are active enhancers. Chromatin is also accessible at promoters, insulators, and regions bound by repressors (*Thurman et al., 2012*; *Kok and Arnosti, 2015*; *Xi et al., 2007*; *Li and Arnosti, 2011*), and hence, enhancer discovery using open chromatin assays has a high false positive rate. Interestingly, chromatin accessibility has been shown to be dynamic across space and time at active developmental enhancers compared to other regulatory elements like promoters (*Reddington et al., 2020*;

*Bozek et al., 2019*; *McKay and Lieb, 2013*), and therefore, dynamic chromatin accessibility has been proposed as an accurate predictor for active enhancers. Thus, in this paper, we used a time-specific and tissue-specific ATAC-seq assay to elucidate the dynamics of open chromatin in space and time in the early *Tribolium* embryos, used the assay to discover a number of active *Tribolium* enhancers, and assessed the association between differential ATAC-seq peak accessibility and enhancer activity.

The second step to understand how enhancers mediate dynamic and wavelike gene expression patterns is to characterize the spatiotemporal dynamics driven by the discovered enhancers. In situ staining of carefully staged embryos can go a long way in characterizing dynamic gene expression patterns. However, salient features of these dynamics can be missed using this method, and a strategy to visualize enhancer activities in live embryos is thus needed. Using fluorescent proteins (FP) as reporters for enhancer activities has been traditionally the method of choice in live imaging studies. Nonetheless, FPs suffer from low degradation rates, which results in averaging out of fast-changing gene expression patterns, rendering them unsuitable for visualizing highly dynamic gene activities. One strategy to tackle this problem is to tag RNAs (*Pichon et al., 2018*), like in MS2 tagging (*Johansson et al., 1997*), where MS2 tandem repeats are inserted within a reporter gene. Upon reporter gene activation, the MS2 repeats are transcribed into stem-loops that readily bind the MS2 virus coat protein (MCP). If MCP-FP fusion proteins are ubiquitously present in the background, they are then recruited at the transcription site in as many numbers as RNA polymerases are actively transcribing the MS2 reporter gene, offering a natural form of signal amplification. This strategy can be used to visualize de novo transcription (*El-Sherif and Levine, 2016*; *Lim et al., 2018*; *Garcia et al., 2013*; *Lucas et al., 2013*; *Bothma et al., 2014*), avoiding the averaging effect of using FPs as reporters. Therefore, to study the dynamics of gene expression waves during embryogenesis, we established an MS2-tagging system in *Tribolium*, and used it to visualize the activities of some of the enhancers we discovered using our enhancer discovery system.

In summary, we established in this paper a framework for enhancer discovery and enhancer activity visualization in both fixed and live embryos in *Tribolium*. First, we assayed the dynamics of open chromatin in space and time in the *Tribolium* embryo using ATAC-seq, and used the assay to discover a number of active enhancers. Of importance to future efforts in that vein, we found that active enhancer regions overlap with chromatin-accessible sites that significantly vary across the AP axis of the embryo. Second, we established an MS2-MCP enhancer reporter system in *Tribolium* to visualize the activity dynamics of discovered enhancers in both fixed and live embryos. Using this enhancer reporter system, we showed that some of the discovered enhancers regulating gap and pair-rule genes feature expression patterns that are in line with the Enhancer Switching model. In particular, we found one enhancer regulating the pair-rule gene *runt* (*run*) that matches the role of a static enhancer, and another enhancer regulating the gap gene *hunchback* (*hb*) that matches the role of a dynamic enhancer. While these data are in line with our Enhancer Switching model, more data is needed as a strong support for the model.

## Results

### Profiling chromatin accessibility landscape along the AP axis of the early *Tribolium* embryo

Genomic regions of increased chromatin accessibility are typically endowed with regulatory activity (*McKay and Lieb, 2013*; *Simon et al., 2012*). At enhancers in particular, chromatin accessibility has been shown to be dynamic across space and time, and so we set out to assay the dynamics of the accessible chromatin landscape in the *Tribolium* embryo. To that end, we dissected the *Tribolium* embryo at the germband stage into three regions across its AP axis (*Figure 2A*): anterior ('a'), middle ('m'), and posterior ('p'), and performed ATAC-seq on each region at two-time points: 23–26 hr after egg-lay (AEL) (hereafter, termed IT23), and 26–29 hr AEL (hereafter, termed IT26) (Materials and methods; see representative embryo from these two stages in *Figure 2A'*). ATAC-seq libraries compromised an average of 1,835,762 unique, high-quality pairs of reads (3.6 X genomic coverage, Materials and methods). Biological replicates of our ATAC-seq libraries were highly similar with a median Spearman's correlation coefficient of 0.875 (*Figure 2—figure supplement 1*), demonstrating the reproducibility of the data.

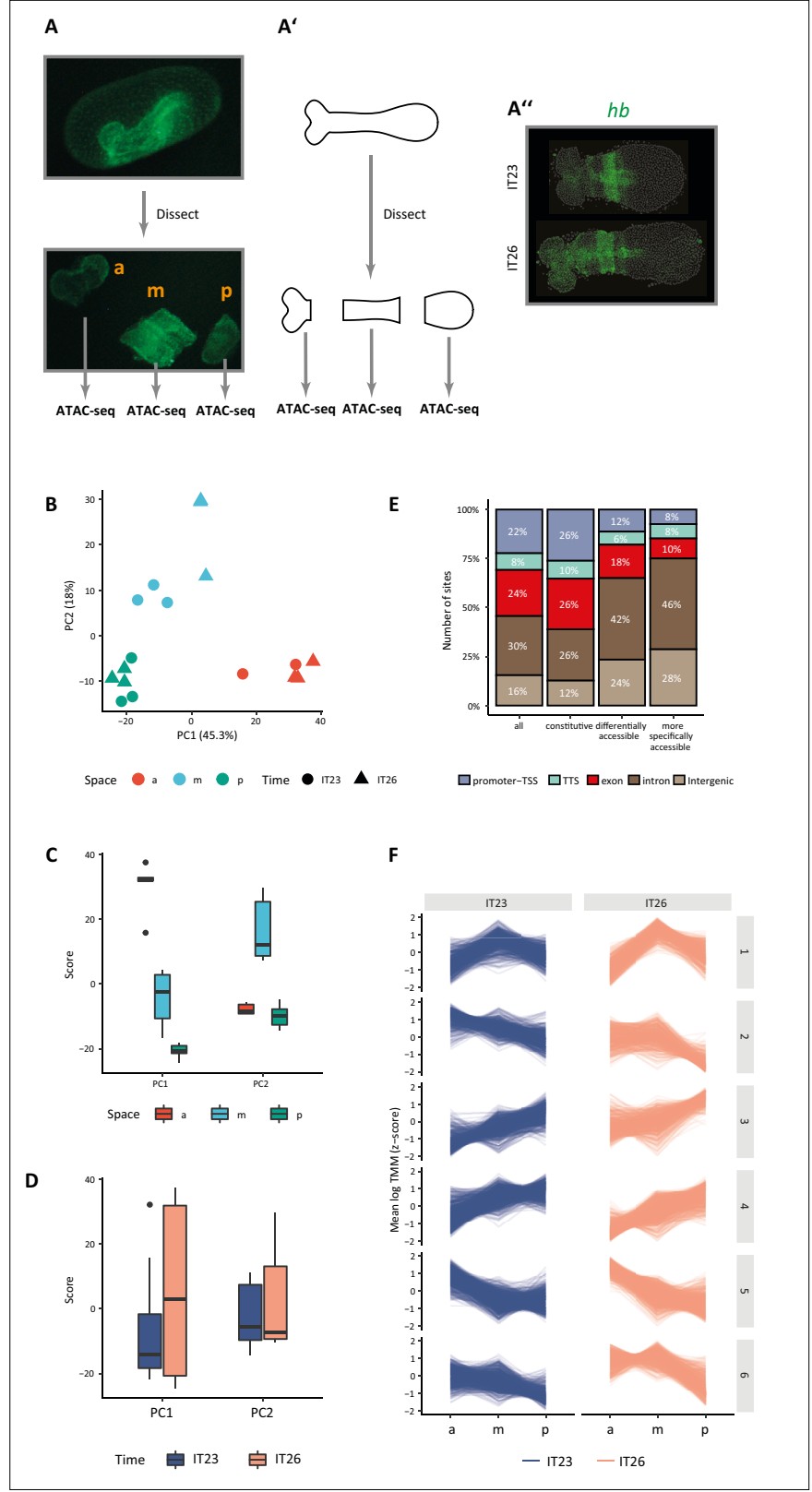

**Figure 2.** Chromatin accessibility dynamics during anterior-posterior (AP) patterning of the early *Tribolium* embryo. (**A**) Embryos 23–26 hr after egg lay (AEL) (IT23) or 26–29 hr AEL (IT26) were dissected and cut into anterior (a), middle (m), and posterior (p) part. Up: nGFP embryo in the eggshell. Below: dissected and cut nGFP embryo. (**A'**) A schematic version of (**A**). (**A"**) Representative *Tribolium* embryo at 23–26 hr AEL (IT23) or 26–29 hr AEL (IT26)

*Figure 2 continued on next page*

*Figure 2 continued*

in situ stained to visualize the expression of the gap gene *hunchback* (*hb*) (green; Hoechst in gray). (**B**) Principal component analysis (PCA) on the accessibility scores of the most highly accessible and variable sites in the dataset. Only the first two principal components (PC) of the data are represented. The first PC explains 45.3% of the variance in the data, and the second PC, 18.0%. (**C, D**) Boxplots showing the scores of PC1 and PC2 by space (**C**) and time (**D**). The thick line indicates the median (2nd quartile), while the box represents the interquartile range (IQR, 1st to 3rd quartiles). Outliers are shown as dots. (**E**) Genomic annotation of different classes of chromatin accessible sites: all (consensus) sites, constitutive sites (i.e. consensus sites that are not differentially accessible), all differentially accessible sites, and most specifically accessible sites (i.e. sites differentially accessible in four or more comparisons). (**F**) K-means clustering of accessibility scores for differentially accessible sites. Accessibility scores have been z-score scaled for each site.

The online version of this article includes the following figure supplement(s) for figure 2:

**Figure supplement 1.** Correlation between Assay for Transposase-Accessible Chromatin with high-throughput sequencing (ATAC-seq) sequencing libraries.

**Figure supplement 2.** Upset plot comparing the number of sites identified in samples corresponding to IT23 and IT26 as well as along the anterior-posterior (AP) axis.

**Figure supplement 3.** Annotation of consensus sites.

**Figure supplement 4.** Analysis of differentially accessible sites.

---

We identified a total of 12,069 chromatin-accessible sites (Materials and methods), with 4017 being specific to one or two particular regions of the germband and 1610 to a given time point (*Figure 2—figure supplement 2*). In agreement with the ability of ATAC-seq to detect distal regulatory elements in the genome (*Buenrostro et al., 2013*), a large proportion of these sites were intergenic or intronic (46%, *Figure 2—figure supplement 3*). Principal component analysis (PCA) of most variable accessible sites (Materials and methods) mainly separated the samples along the AP axis of the embryo (*Figure 2B–D*). Among all 12,069 sites, 3106 (26% of the accessible genome) were differentially accessible when compared between different regions along the AP axis and/or time point (Materials and methods). For 1049 of those sites, changes in accessibility were observed in four or more comparisons, indicating more intricate and, generally, specific patterns of accessibility (*Figure 2—figure supplement 4*). Remarkably, while 62% of the constitutively accessible sites corresponded to promoters and gene bodies, 66% of the differentially accessible sites were intergenic or intronic, and this proportion was even higher (74%) among the differentially accessible sites with more specific patterns of accessibility (*Figure 2E*), suggesting that spatiotemporal control of transcription in the early *Tribolium* embryo is largely mediated by enhancers as opposed to promoters. When comparing accessibility along the AP axis at a particular time point, 2109 sites were differentially accessible, the majority of them along the AP axis at IT26. In addition, only 132 sites were differentially accessible between IT26 and IT23 at the same portion of the embryo (*Figure 2—figure supplement 4B and C*). To gain a better insight into the spatial and temporal dynamics of chromatin accessibility, we clustered all differentially accessible sites across the germband regions and time points (*Figure 2F*). Almost half of the sites were either not accessible in the anterior region of the embryo but accessible in the middle and posterior regions (cluster 4, with 756 sites), or not accessible in the middle and posterior regions of the embryo but accessible in the anterior region (cluster 5, with 698 sites).

Together, our findings indicate that changes in chromatin accessibility in *Tribolium* at this developmental stage are primarily associated with space rather than time, and are particularly evident when comparing the anterior part of the germband to the middle and posterior parts. Furthermore, our data are in line with observations made by other authors (*Cusanovich et al., 2018*; *Bozek and Gompel, 2020*) that suggests that dynamically accessible sites are especially likely to be associated with enhancer activity, laying the foundation for a promising enhancer prediction strategy based on differential ATAC-seq peak analysis. Before assessing this proposition, however, we set out to establish an enhancer reporter system to validate the activity of predicted enhancers, and analyze their transcriptional dynamics.

## Establishing an MS2-MCP enhancer reporter system to visualize enhancer activity in fixed and live *Tribolium* embryos

An enhancer reporter system has been previously established in *Tribolium* using *mCherry* as a reporter gene (*Lai et al., 2018*). However, long half-lives of *mCherry* mRNA and proteins could average out fast transcriptional dynamics, precluding the analysis of gene expression waves. To circumvent this, we created a *Tribolium* enhancer reporter system capable of visualizing de novo transcription in both fixed and live embryos. Our enhancer reporter system is composed of the gene *yellow*, which has a long intron (2.7 kb). Visualizing intronic transcription of the reporter gene *yellow* using in situ staining in fixed embryos enables the detection of de novo transcription, and has been routinely used to analyze fast transcriptional dynamics in enhancer reporter experiments in *Drosophila* (*Perry et al., 2011*). To visualize de novo transcription in live *Tribolium* embryos, we set out to (*i*) modify the *yellow* reporter gene to allow for MS2 tagging, and (*ii*) create a *Tribolium* transgenic line with ubiquitous expression of an MCP-FP fusion. To that end, we created two piggyBac reporter constructs: 'enhancer >MS2-*yellow*' (*Figure 3A*) and 'aTub >MCP-mEmerald' (*Figure 3B*).

This system is capable of visualizing enhancer activity both in fixed and live embryos. To visualize aggregate enhancer activity in fixed embryos, *yellow* gene expression is visualized using in situ staining in embryos carrying the enhancer >MS2-*yellow* construct. To visualize de novo transcription in fixed embryos, an in situ probe against *yellow* intron is used instead. To visualize de novo transcriptional activity of an enhancer in live embryos, a male beetle carrying the enhancer >MS2-*yellow* construct is crossed with a female beetle carrying the aTub >MCP-mEmerald construct. If active, the enhancer should drive the expression of the MS2-*yellow* reporter in the progeny. The transcribed MS2 loops would then recruit aTub >MCP-mEmerald fusion proteins at the transcription site of the reporter gene, enriching the mEmerald fluorescent signal against the weak mEmerald background.

Via piggyBac transgenesis, we successfully generated a transgenic beetle line carrying the MCP-mEmerald construct, in which a ubiquitous mEmerald fluorescence is detected (*Figure 3C*). We then sought to test our enhancer >MS2-*yellow* reporter system, using a previously discovered *Tribolium* enhancer, hbA, that regulates the *Tribolium* gap gene *hunchback* (*hb*) (*Lai et al., 2018*). *Hb* is expressed in multiple domains in the early *Tribolium* embryo: in the serosa, in an anterior domain, in a secondary posterior domain (shown in orange, blue, and purple, respectively in *Figure 3D*), and in the nervous system (not shown). Enhancer hbA drives the anterior expression of *Tribolium hb* (*Lai et al., 2018*) (blue in *Figure 3D*). Via piggyBac transgenesis, we successfully generated a transgenic beetle line carrying the hbA >MS2-*yellow* construct. Examining *yellow* expression using in situ hybridization chain reaction (HCR) (*Choi et al., 2018*) in early hbA >MS2-*yellow* embryos using both exonic (*Figure 3E and F*) and intronic (*Figure 3F*) probes, we confirmed that the *yellow* expression in hbA >MS2-*yellow* line is similar to the *mCherry* expression in a previously tested hbA >*mCherry* line (*Lai et al., 2018*).

To test the live imaging capability of our MS2-MCP system, we crossed the hbA >MS2-*yellow* and the aTub >MCP-mEmerald lines. Imaging early embryos of the progeny (hbA >MS2-*yellow*; aTub >MCP-mEmerald) (*Video 1*; *Figure 3G*), we observed weak and diffuse mEmerald signal within the nuclei, and bright puncta at a rate of at most one punctum per nucleus. The bright mEmerald puncta are distributed along the AP axis initially as a cap that eventually refines into a stripe (*Figure 3—figure supplement 1*), resembling the *yellow* expression of the hbA enhancer reporter visualized using in situ HCR staining (*Figure 3E and F*). We conclude, therefore, that such bright mEmerald puncta are mEmerald enrichments at transcribed MS2 loops, reflecting the de novo transcription driven by the hbA enhancer. Hence, both individual nuclei of the early *Tribolium* embryo and de novo transcription driven by the hbA enhancer can be visualized and detected in a single cross of hbA >MS2-*yellow* line and the MCP-mEmerald line, confirming our success in establishing an MS2-MCP enhancer reporter system that is capable of visualizing enhancer activity in live *Tribolium* embryos.

## Assessing the association between differential accessibility and enhancer activity

We then sought to use our enhancer reporter system to test putative enhancers suggested by our ATAC-seq analysis. In selecting a set of putative enhancers to test, we restricted our analysis to genomic regions around three genes, all involved in AP patterning in *Tribolium*: the gap genes *hb* (*Figure 4A*) and *Kr* (*Figure 4—figure supplement 1A*) as well as the pair-rule gene *runt* (*run*)

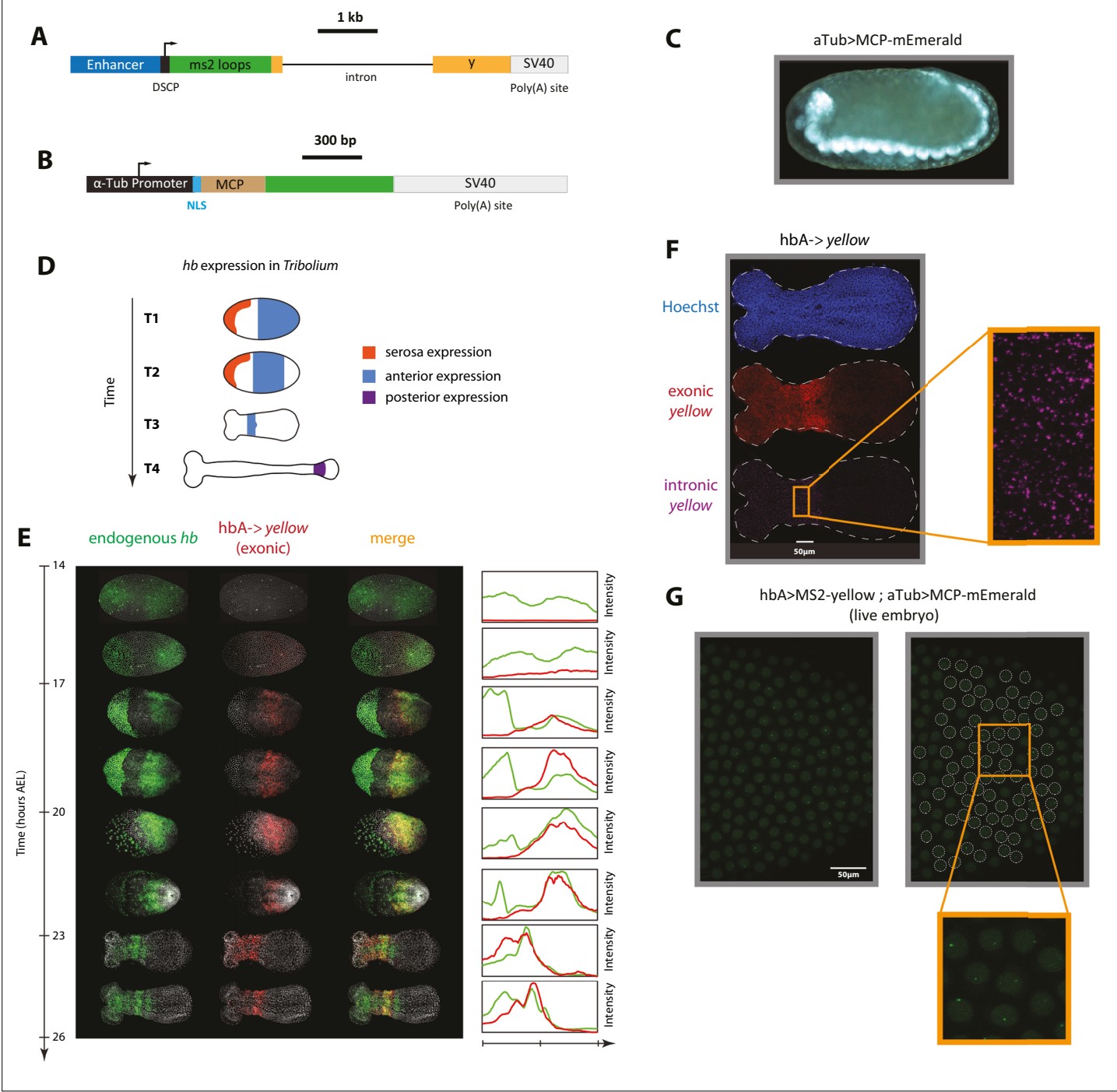

**Figure 3.** An MS2-MCP enhancer reporter system to visualize enhancer activity in fixed and live *Tribolium* embryos tested using the *Tribolium hb* enhancer hbA. (**A**) Our enhancer reporter construct: An enhancer of interest is placed upstream of a Synthetic Core Promoter (DSCP), followed by 24 tandem repeats of MS2 stem-loops, the gene *yellow*, then an SV40 poly(A) tail. (**B**) The aTub >MCP-mEmerald construct: ubiquitous alpha-tubulin promoter was placed upstream of an nuclear localization signal (NLS) and an MCP-mEmerald fusion, followed by an SV40 poly(A) tail. (**C**) Overview image of an aTub >MCP-mEmerald embryo at the germband stage. (**D**) A schematic showing *hb* expression in *Tribolium*. In the early blastoderm, *hunchback* (*hb*) is expressed in the serosa (orange) and as a cap in the posterior half of the embryo (blue) (T1) that eventually resolves into an expression band in the anterior (blue in T2 and T3). Later during the germband stage (T4), the anterior expression (blue) fades and a new *hb* expression arises in the posterior (purple). (**E**) Spatiotemporal dynamics of endogenous *hb* (green) and reporter gene *yellow* (red) expression in hbA->*yellow Tribolium* embryos. Left panel: Time-staged embryos from 14 to 26 hr after egg lay (AEL) at 24 °C, in which mRNA transcripts (*hb*: green, *yellow*: red) were visualized using in situ hybridization chain reaction (HCR) staining. Nuclear staining (Hoechst) is in gray. Right panel: Fluorescence signal along the dorsal-ventral axis was summed up to generate intensity distribution plots along the anterior-posterior (AP) axis. (**F**) Detection of de novo transcription via in situ HCR staining

*Figure 3 continued on next page*

*Figure 3 continued*

of intronic *yellow*. Nuclear staining (Hoechst): blue; exonic *yellow* (*yellow* mRNA): red; intronic *yellow*: purple. Embryo outline is shown in a dashed line. (**G**) Live imaging snapshot of a hbA >MS2-*yellow*; aTub >MCP-mEmerlad *Tribolium* embryo. Diffuse mEmerald signal is observed in nuclei (outlined in a white dashed line). mEmerald fluorescence is enriched at transcription sites (bright puncta: MS2-MCP signal). In all embryos shown: posterior to the right.

The online version of this article includes the following figure supplement(s) for figure 3:

**Figure supplement 1.** Analysis of hbA enhancer activity on a fixed embryo.

(***Figure 4B***). Candidate enhancer regions were chosen based on the presence of accessible sites in any region along the AP axis and/or time point (Materials and methods), whether or not they were differentially accessible.

Out of nine tested reporters, four successfully drove *yellow* expression in the early *Tribolium* embryo (***Figure 4C***). While enhancer hbA drove an expression that overlaps with *hb* anterior expression, enhancer hbB drove an expression that overlaps with *hb* expression in the serosa (compare hbA and hbB activities in ***Figure 4C***; see ***Figure 3D*** for the constituents of *hb* expression in *Tribolium*). Enhancer runA drove an expression that partially overlaps the endogenous *run* expression in the nervous system during the late germband stage (runA in ***Figure 4C***). Enhancer runB drove a striped expression that overlaps endogenous *run* expression in the ectoderm, but neither the striped *run* expression in the mesoderm, nor the nervous system expression (runB in ***Figure 4C***). In addition, runB drove an expression in the head lobes (asterisks in ***Figure 4C***) that is missing in the endogenous *run* expression, possibly due to some missing repressors flanking the selected DNA segment for runB.

We then determined whether there is any association between differential accessibility and enhancer activity using our tested enhancer constructs, augmented with previously published *Tribolium* enhancers that regulate the genes *single-minded* (*sim*) and *short gastrulation* (*sog*) (***Cande et al., 2009***; ***Figure 4—figure supplement 1B and C***). In total, we analyzed 11 enhancer constructs, among which six (54.5%) were active and five (45.5%) were not. We found that out of the six active constructs, five (83%) overlapped differentially accessible sites, while one (17%) overlapped a site that was not differentially accessible. In contrast, out of the five non-active constructs, three (60%) overlapped sites that were differentially accessible, while the remaining two (40%) overlapped sites that were not differentially accessible (***Figure 4D***; see ***Figure 4—figure supplement 1*** for details). Therefore, about 60% of analyzed differentially accessible sites were associated with active enhancers whereas only 30% of analyzed constitutively accessible sites were associated with active enhancer regions (***Figure 4E***). Although the sample size is small, the trend is in line with observations in other model systems (***Cusanovich et al., 2018***; ***Bozek and Gompel, 2020***) suggesting that differential accessibility is associated with enhancer activity.

## Testing the plausibility of the Enhancer Switching model

Next, we set out to test the plausibility of the Enhancer Switching model by examining the activity dynamics of some of the discovered enhancers. The model predicts that for a gene involved in generating gene expression waves, there exist two enhancers: (*i*) a 'dynamic enhancer' responsible for initiating the wave, and (*ii*) a 'static

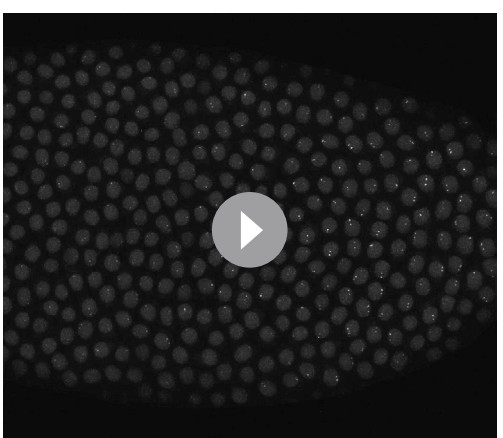

**Video 1.** MS2 live imaging of hbA enhancer reporter. Live imaging of a 'hbA >MS2-*yellow*; aTub >MCP-mEmerlad' *Tribolium* embryo during the blastoderm stage. Nuclear localization signal (NLS) signal within the aTub >MCP-mEmerald construct mediates a weak and diffuse mEmerald signal within nuclei. Upon transcription, MS2 loops within the hbA >MS2-*yellow* construct recruit MCP-mEmerald fusion proteins at transcription sites, resulting in mEmerald bright puncta. Here bright mEmerald puncta are observed throughout the posterior end of the blastoderm, reflecting transcriptional activity of enhancer hbA in the early *Tribolium* embryo. Posterior to the right.
https://elifesciences.org/articles/84969/figures#video1

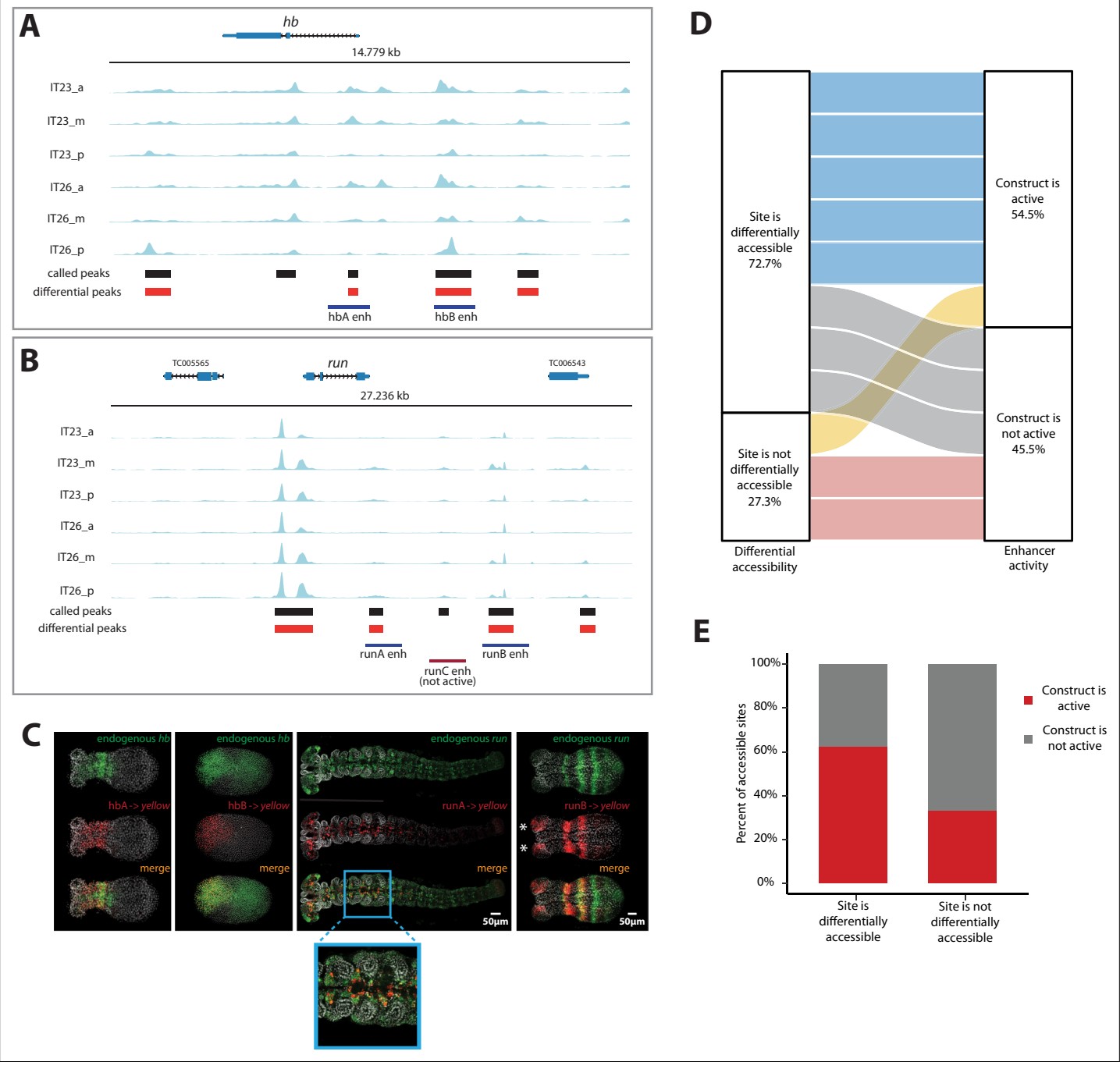

**Figure 4.** Correlation of enhancer activity with differential accessibility. (**A, B**) The Assay for Transposase-Accessible Chromatin (ATAC) profiles (two-time points (IT23, IT26) with three embryo regions (a, m, p) per time point) for hunchback (*hb*) (**A**) and *run* (**B**). Tested enhancer regions at these loci are shown as boxes underneath the ATAC profiles. Differential accessible sites match well with active enhancer regions (purple boxes; red box: not active enhancer construct). ATAC tracks were created with pyGenomeTracks. (**C**) Enhancer reporter constructs for active enhancer regions are shown in (**A**) and (**B**), in which mRNA transcripts were visualized using in situ hybridization chain reaction (HCR) staining (endogenous gene expression: green, reporter gene expression: red, merge: yellow). Nuclear staining (Hoechst) is in gray. hbA drives reporter gene expression in a stripe (embryo in germband stage shown). hbB drives reporter gene expression in the serosa (embryo in blastoderm stage shown). runA drives reporter gene expression in a subset of the endogenous *run* CNS expression (embryo in late germband stage shown). runB drives reporter gene expression in stripes outside of the most posterior part of the embryo (embryo in germband stage shown). Posterior to the right. (**D**) The correlation between differential accessibility and construct activity was determined. Eleven enhancer constructs were analyzed: 54.5% of constructs (six constructs) were active and 45.5% of constructs were not active (five constructs). Five out of six active constructs are associated with sites that are differentially accessible, while one active construct overlaps with a site that is not differentially accessible. Two out of five not active constructs match sites that are not differentially accessible, while the remaining three not active

*Figure 4 continued on next page*

*Figure 4 continued*

constructs are associated with sites that are differentially accessible (see *Figure 4—figure supplement 1* for details). (**E**) Enhancer prediction efficiency of our enhancer prediction method based on differential peak analysis. Same enhancer constructs were analyzed as in (**D**). About 60% of analyzed differential peaks were associated with active enhancer construct regions whereas in about 40% of analyzed cases, differential peaks could be found at not active enhancer construct regions. In contrast, about 70% of analyzed non-differential peaks were associated with not active enhancer construct regions. About 30% of analyzed non-differential peaks are associated with active enhancer construct regions.

The online version of this article includes the following figure supplement(s) for figure 4:

**Figure supplement 1.** Genomic tracks of analyzed enhancer reporter constructs.

enhancer' responsible for arresting the wave into a stable gene expression domain(s) (*Figure 1C and D*).

Among the discovered enhancers in *Tribolium*, two enhancers are potentially involved in generating gene expression waves: hbA and runB. The expression patterns driven by both enhancers overlap with the expression waves of their corresponding genes: enhancer runB with the periodic waves of the pair-rule gene *run*, and enhancer hbA with the non-periodic wave of the gap gene *hb*. To test if the spatio-temporal dynamics driven by these enhancers conform with some of the predictions of the Enhancer Switching model, we first ran simulations of the model and used them to carefully analyze model predictions. Then, we used our enhancer reporter system to track the enhancer activity dynamics of runB and hbA in space and time using in situ HCR staining in carefully staged fixed embryos as well as using our MS2-MCP system in live *Tribolium* embryos. Finally, we compared our model predictions with the observed enhancer activity dynamics.

## Careful examination of the predictions of the Enhancer Switching model

To carefully analyze the predictions of the Enhancer Switching model in space and time, we ran a simulation (*Video 2*) of a 3-genes realization of the periodic version of the model (*Figure 1C*, where an oscillator is used as a dynamic module) (see Materials and methods). Carefully analyzing model outputs for the total activity of constituent genes, static enhancer reporters, and dynamic enhancer reporters revealed two characteristics of the spatiotemporal dynamics of their activities (*Figure 5A*). First, endogenous genes and reporters of their dynamic and static enhancers are all expressed in waves that propagate from posterior to anterior (*Figure 5A*). Second, expression patterns driven by dynamic enhancers progressively decrease in the posterior-to-anterior direction, matching the progressive decrease of the speed regulator concentration (*Figure 5A*). On the other hand, expression patterns driven by static enhancers progressively increase in the posterior-to-anterior direction, opposite to the direction of the increase of the speed regulator concentration (*Figure 5A*). This is a natural consequence of the activating vs repressing effect of the speed regulator on dynamic vs static enhancers, respectively (*Figure 1C*).

However, a minor complication arises when one considers a more realistic instantiation of the Enhancer Switching model. In our simulation of the model presented in *Figure 5A* and in our simulations presented in previous publications (*Zhu et al., 2017*; *Boos et al., 2018*; *Rudolf et al., 2020*), we assumed that the stabilized gene expression domains at the anterior remain stable indefinitely (*Figure 5A*). However, we observe experimentally that such astable phase is transient, after which gene expression domains gradually fade (notice the progressive fading of the first *run* stripe after its stabilization at the anterior in *Figure 6A and B*). This effect can be implemented computationally

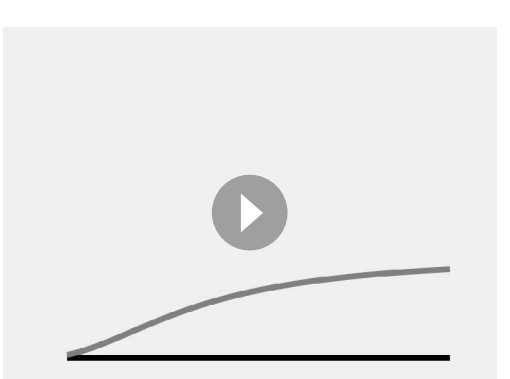

**Video 2.** Simulation of the Enhancer Switching Model with strong static enhancer activity. Shown are the outputs of a computer simulation of the Enhancer Switching model with strong static enhancer activity. Activity dynamics of reporter genes driven by the dynamic and static enhancers are shown in yellow and black, respectively. Activity dynamics of endogenous gene expression driven by both the dynamic and static enhancers are shown in green. Speed regulator gradient is shown in gray.

https://elifesciences.org/articles/84969/figures#video2

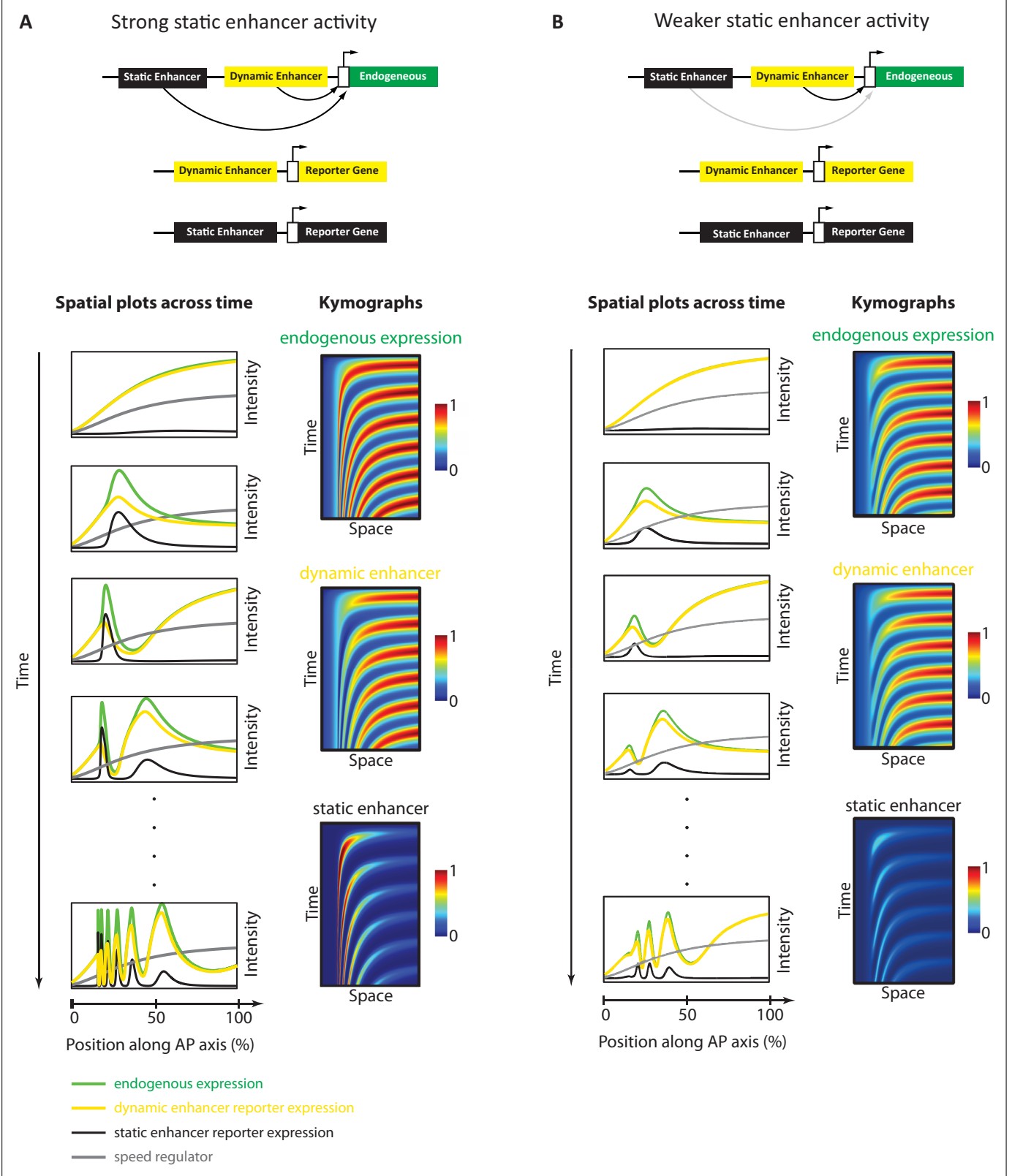

**Figure 5.** Simulation of the Enhancer Switching model with different static enhancer strengths. Shown are simulation outputs of the Enhancer Switching model for a reporter gene driven by dynamic (yellow) or static (black) enhancers, as well as an endogenous gene driven by both dynamic and static enhancers (green). Two versions of the model were simulated and contrasted: with strong (**A**) vs weak (**B**) static enhancer activity. (**A**) Model simulation with strong static enhancer activity. Each wave of the endogenous gene expression follows first the dynamics of the dynamic enhancer and switches

*Figure 5 continued on next page*

*Figure 5 continued*
along space (in the tapering direction of the speed regulator, shown in gray) and time to the dynamics of the static enhancer to form a stable expression domain. (**B**) Model simulation with a weaker static enhancer: dynamic enhancer activity resembles endogenous gene expression pattern. Left panels: spatial plots across time. Right panels: Kymographs.

The online version of this article includes the following figure supplement(s) for figure 5:

**Figure supplement 1.** Simulations with different static enhancer strengths.

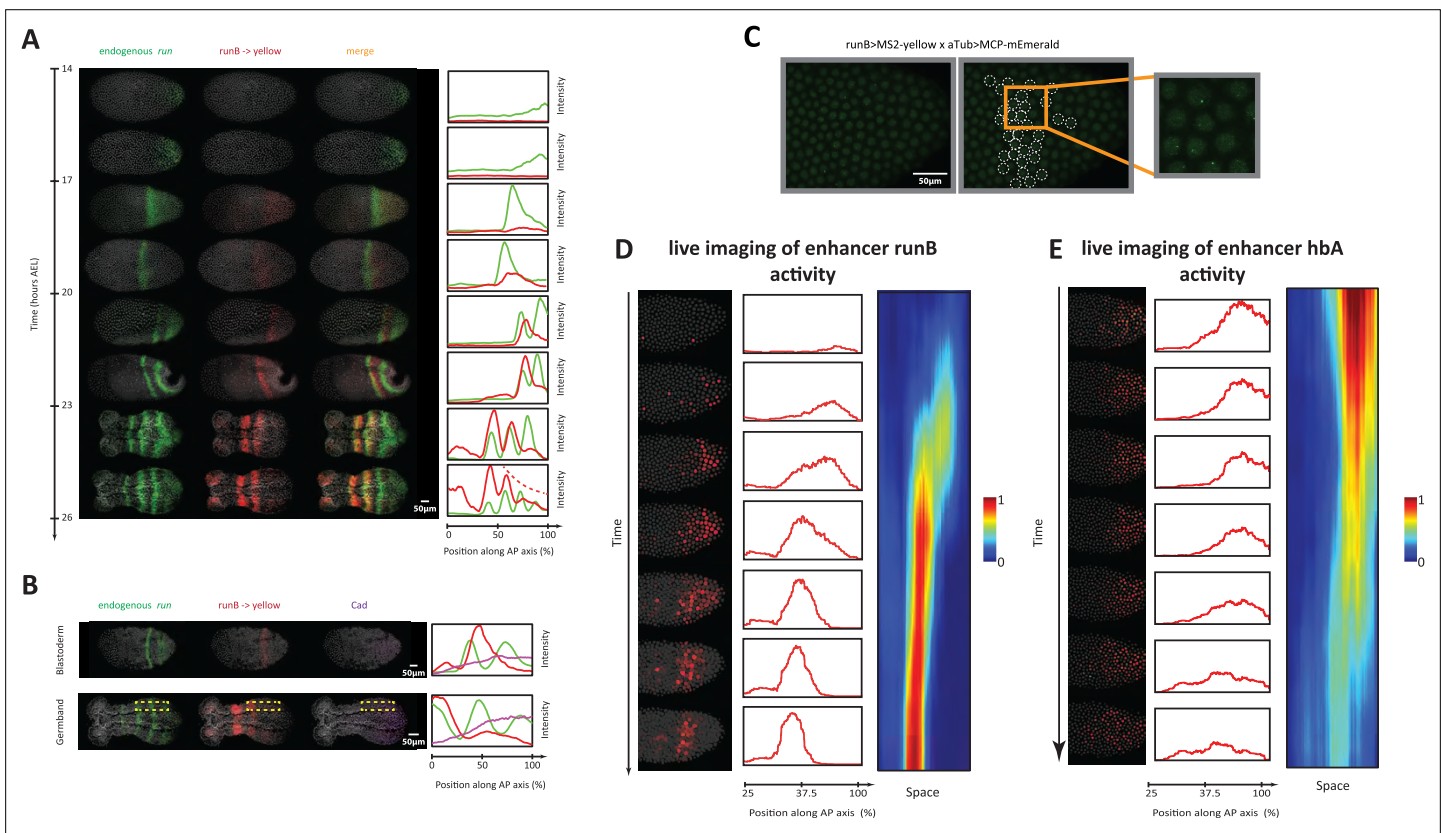

**Figure 6.** Analysis of enhancer activity dynamics using the MS2-MCP live imaging system. (**A**) Shown are spatiotemporal dynamics of endogenous *run* (green) and the reporter gene *yellow* (red) expression in runB->*yellow* embryos. Left panel: Time-staged embryos from 14 to 26 hr after egg lay (AEL) at 24 °C, in which mRNA transcripts (*run*: green, *yellow*: red) were visualized using in situ hybridization chain reaction (HCR) staining. Nuclear staining (Hoechst) is in gray. Right panel: Intensity distribution plots along the AP axis. Both *run* and runB->*yellow* are expressed in waves that propagate from posterior to anterior. runB->*yellow* expression wave, however, starts weak posteriorly and progressively increases in strength as it propagates towards the anterior, until it eventually overlaps with the stabilized *run* stripes anteriorly. (**B**) Left panel: in situ HCR staining for endogenous *run* (green) and reporter gene *yellow* expression (red) combined with antibody staining for Cad proteins (purple) in a runB->*yellow* blastoderm embryo (upper row) and a runB->*yellow* germband embryos (lower row). Nuclear staining (Hoechst) is in gray. Right panel: Intensity distribution plots along the AP axis for *yellow* expression in a whole *Tribolium* blastoderm (upper panel), and within the region indicated in dashed yellow in a *Tribolium* germband. Cad forms a posterior-to-anterior gradient in both balstoderm and germband embryos. runB activity increases progressively as Cad concentration drops towards anterior. (**C**) Live imaging snapshot of a runB >MS2-*yellow*; aTub >MCP-mEmerlad *Tribolium* embryo. Diffuse mEmerald signal is observed in nuclei. mEmerald fluorescence is enriched at transcription sites (bright puncta: MS2-MCP signal). Left panel: original snap shot; Middle: nuclei that exhibit MS2-MCP signal are outlined in white circles; Right: A close-up to nuclei with MS2-MCP signal. (**D, E**) Tracking estimated mRNA activity driven by runB (**D**) and hbA (**E**). Left panels (in both (**D**) and (**E**)): Snapshots across time from live embryo movies in which nuclei (shown in gray) are tracked and MS2 signals are averaged over time (using a moving average filter with a length of 10 movie frames) to estimate mRNA activity (shown in red). Middle panel: Intensity distribution plots along space for representative images in left panel. Right panel: A kymograph showing estimated mRNA activities of enhancer reporters across space and time. In all embryos shown: posterior to the right.

The online version of this article includes the following figure supplement(s) for figure 6:

**Figure supplement 1.** Visualizing runB >yellow expression waves in the germband using exonic vs intronic probes.

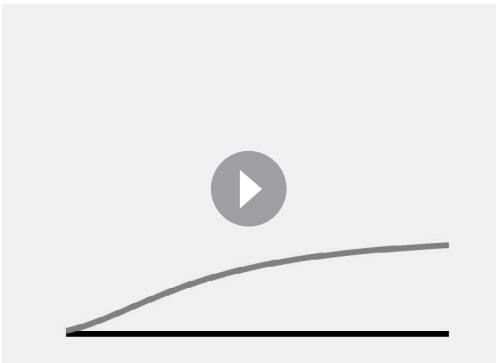

**Video 3.** Simulation of the Enhancer Switching Model with weak static enhancer activity. Shown are the outputs of a computer simulation of the Enhancer Switching model with weak static enhancer activity. Activity dynamics of reporter genes driven by the dynamic and static enhancers are shown in yellow and black, respectively. Activity dynamics of endogenous gene expression driven by both the dynamic and static enhancers are shown in green. Speed regulator gradient is shown in gray.

https://elifesciences.org/articles/84969/figures#video3

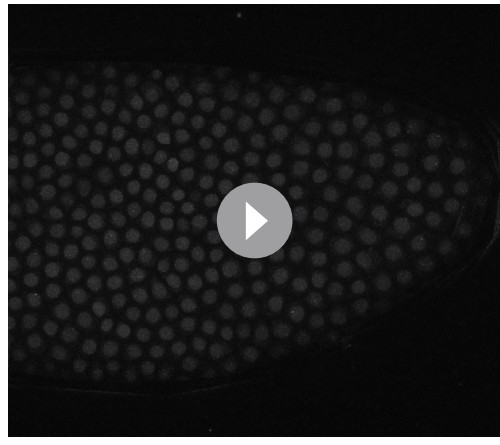

**Video 4.** MS2 live imaging of runB enhancer reporter. Live imaging of a 'runB >MS2-*yellow*; aTub >MCP-mEmerlad' *Tribolium* embryo during the blastoderm stage. Nuclear localization signal (NLS) signal within the aTub >MCP-mEmerald construct mediates a weak and diffuse mEmerald signal within nuclei. Upon transcription, MS2 loops within the runB >MS2-*yellow* construct recruit MCP-mEmerald fusion proteins at transcription sites, resulting in mEmerald bright puncta. Here bright mEmerald puncta are observed initially to be distributed as a posterior cap that eventually propagates towards the anterior to form a stable band. Posterior to the right.

https://elifesciences.org/articles/84969/figures#video4

by reducing the strength of the static enhancers (*Figure 5B*, *Video 3*, *Figure 5—figure supplement 1*). In this case, the expression driven by a dynamic enhancer is very similar to the total expression of the gene expression wave: both arise maximally at the posterior and gradually fade as they propagate towards the anterior (*Figure 5B*). On the other hand, the expression wave driven by the static enhancer remains unique, as it, for the most part, increases in the direction of its propagation (until it eventually fades; *Figure 5B*). This means that while it is easy to identify a static enhancer from its enhancer reporter activity, it is not as simple to identify a dynamic enhancer. In particular, an enhancer that drives an expression that arises maximally at the posterior and gradually fades as it propagates towards the anterior might be either a dynamic enhancer or simply an enhancer that drives the entirety of the gene expression wave.

## Examining the activity dynamics of enhancer runB using in situ HCR staining

To test the predictions of the Enhancer Switching model, we started by examining the spatiotemporal dynamics of one of the primary pair-rule genes, *run*, simultaneously with those of one of its enhancers that we discovered in this study, runB, using HCR in carefully staged embryos (*Figure 6A*). Endogenous *run* expression (green in *Figure 6A*) periodically arises from the posterior and gradually propagates towards the anterior, forming a stable striped expression (that eventually fades). Concomitantly, runB >*yellow* is expressed as well in a periodic wave that propagates from posterior to anterior (red in *Figure 6A*). However, in contrast to endogenous *run* expression, the expression wave of runB >*yellow* appears weakly in the posterior and gradually strengthens as it propagates towards anterior. These observations are in line with the predicted enhancer dynamics of our model, in which runB acts as a static enhancer for *run*. First, runB drives gene expression waves that propagate from posterior to anterior. Second, runB activity progressively increases in the posterior-to-anterior direction, corresponding inversely with the progressive drop of concentration of the Caudal (Cad) gradient (*Figure 6B*), which has been suggested to act as a speed regulator for pair-rule and gap genes in *Tribolium* (*El-Sherif et al., 2014*; *Zhu et al., 2017*), and more generally, an evolutionary conserved posterior determinant in arthropods (along with Wnt) (*Auman et al., 2017*; *Bolognesi et al., 2008a*; *Schönauer et al., 2016*; *Bolognesi et al., 2008b*; *Copf et al., 2004*; *McGregor et al., 2008*; *Novikova et al., 2020*; *Miyawaki et al., 2004*).

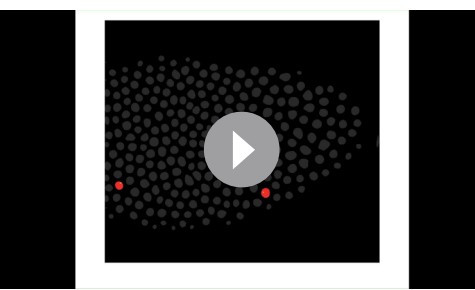

**Video 5.** Estimated mRNA transcription driven by enhancer runB in the early *Tribolium* embryo. Shown is a live imaging movie of a 'runB >MS2-*yellow*; aTub >MCP-mEmerlad' embryo (same as in *Video 4*) computationally processed to show an estimation of accumulated mRNA abundance driven by enhancer runB (red) as well as MS2-mEmerald signal (reflecting de novo transcription; green). Posterior to the right.
https://elifesciences.org/articles/84969/figures#video5

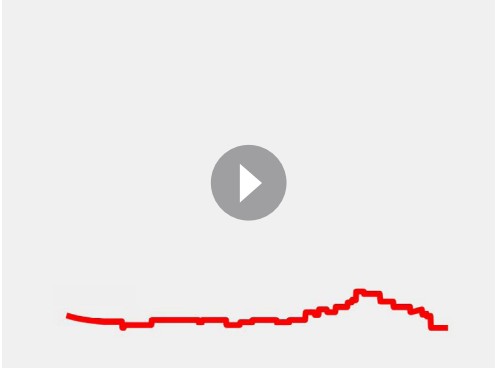

**Video 6.** Plots of estimated mRNA transcription dynamics driven by enhancer runB across space and time. Shown is a dorsoventral projection of a tracked spatiotemporal activity of enhancer runB (same embryo as in *Video 4*). Horizontal axis: anterior-posterior (AP) axis; posterior to the right.
https://elifesciences.org/articles/84969/figures#video6

We notice, however, that the wave dynamics of runB >*yellow* expression are less discernible in posterior germbands (see *Figure 6A* and 23–26 hr AEL). We wondered if this is due to the long degradation delays of *yellow* mRNAs. In line with this possibility, we noticed that mature runB >*yellow* stripes at anterior germbands are more stable and long-lived than those of endogenous *run* (*Figure 6—figure supplement 1*). To circumvent this, we examined runB->*yellow* expression using an intronic probe against *yellow*, and indeed found that the intronic expression of runB->*yellow* is both discernible in the posterior germband and in line with endogenous *run* expression anteriorly (*Figure 6—figure supplement 1*). This shows that, indeed, the degradation delay of the reporter gene *yellow* is larger than that of an endogenous *run*, leading to averaging out of *run* expression wave dynamics, a problem that can be alleviated using intronic in situ staining.

## Examining the activity dynamics of enhancer runB using live imaging

To verify that runB indeed drives expression waves that propagate from posterior to anterior, we performed a live imaging analysis of runB activity using our MS2-MCP system in *Tribolium*. Crossing runB enhancer reporter line (runB >MS2-*yellow*) with our aTub >MCP-mEmerlad line, and imaging early embryos of the progeny, we observed bright mEmerald puncta distributed along the AP axis as a stripe (*Video 4*; *Figure 6C*), an expression that resembles that of *yellow* in the same reporter line visualized using in situ HCR staining (compare *Figure 6A and C*). To characterize the spatiotemporal activity of runB enhancer, circumventing the ambiguity introduced by nuclear and cellular flow, we developed a computational strategy to: (*i*) track the nuclei of the early live *Tribolium* embryo, (*ii*) detect MS2 puncta, and (*iii*) associate the detected MS2 puncta to corresponding nuclei (Materials and methods). Furthermore, to smoothen out the highly stochastic expression of de novo transcription, we applied a moving average window to the MS2 signal across time to estimate an accumulated activity of the runB

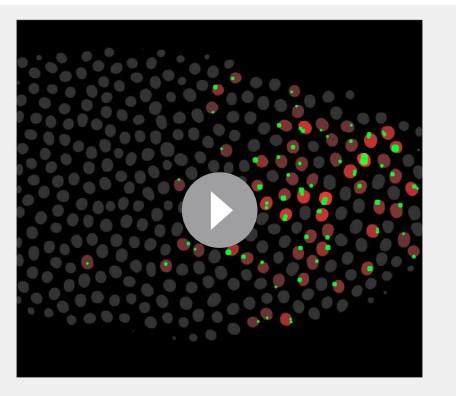

**Video 7.** Estimated mRNA transcription driven by enhancer hbA in the early *Tribolium* embryo. Shown is a live imaging movie of a 'hbA >MS2-*yellow*; aTub >MCP-mEmerlad' embryo (same as in *Video 4*) computationally processed to show an estimation of accumulated mRNA abundance driven by enhancer hbA (red) as well as MS2-mEmerald signal (reflecting de novo transcription; green). Posterior to the right.
https://elifesciences.org/articles/84969/figures#video7

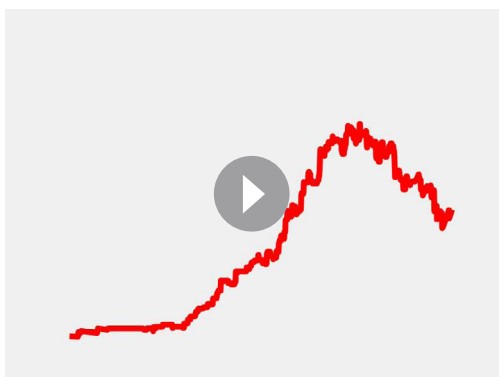

**Video 8.** Plots of estimated mRNA transcription dynamics driven by enhancer hbA across space and time. Shown is a dorsoventral projection of a tracked spatiotemporal activity of enhancer hbA (same embryo as in **Videos 1 and 7**). Horizontal axis: anterior-posterior (AP) axis; posterior to the right.
https://elifesciences.org/articles/84969/figures#video8

enhancer (**Video 5**; Materials and methods). Tracking the accumulated activity in space, after discounting nuclear/cellular flow, revealed that runB indeed drives a wave of activity that progressively increase in strength as it propagates from posterior to anterior (**Video 6**; **Figure 6D**), fitting the role of a 'static enhancer' within our Enhancer Switching model. However, this merely shows that runB activity dynamics are consistent with our model, but is still far from strongly supporting the model (more on that in the Discussion).

## Examining the activity dynamics of enhancer hbA using live imaging

Gap genes are expressed as well in waves in the *Tribolium* embryo, albeit in a non-periodic fashion. Gap gene waves are initialized by a pulse of *hb* expression that arises in the posterior of the blastoderm at 14 hr AEL, that eventually propagates towards the anterior, and clears from the posterior, forming a stripe of *hb* expression at the anterior part of the embryo (**Figure 3D and E**).

Similar to our analysis of runB enhancer, we used our MS2-MCP system to estimate the accumulated mRNA signal driven by enhancer hbA, and tracked it in space and time in live *Tribolium* embryos (**Video 7**, **Video 8**). We found that enhancer hbA drives a wave of activity that propagates from posterior to anterior (**Figure 6E**). In contrast to enhancer runB, however, we found that enhancer hbA drives strong expression in the posterior that weakens as it propagates towards the anterior (compare **Figure 6D and E**). As concluded by our simulations of the Enhancer Switching model (**Figure 5B**), this indicates that hbA either drives the entirety of *hb* expression, or acts as a dynamic enhancer within the Enhancer Switching model. Again, this merely shows that hbA activity dynamics are consistent with our model, but is still far from strongly supporting it.

## Discussion

In this paper, we established a framework for enhancer discovery in *Tribolium* using tissue- and time-specific ATAC-seq (**Figure 2**). We showed that differential accessible site analysis across space and time yields a sizeable increase in enhancer prediction accuracy (**Figure 4**). We also developed an enhancer reporter system in *Tribolium* able to visualize dynamic transcriptional activities in both fixed and live embryos (**Figure 3**). Both our enhancer discovery and activity visualization systems are efforts to establish the AP patterning in *Tribolium* as a model system for studying dynamic gene expression patterns, especially gene expression waves, a phenomenon commonly observed during embryonic development (**Diaz-Cuadros et al., 2021**; **Di Talia and Vergassola, 2022**; **Bailles et al., 2022**). Although our experimental framework is suitable for exploring how enhancers mediate dynamic gene expression in an unbiased fashion, we set in this work to test the plausibility of a specific model: the 'Enhancer Switching' model (**Figure 1**), a scheme some of the authors have recently suggested (**Zhu et al., 2017**; **Kuhlmann and El-Sherif, 2018**) to elucidate how gene expression waves are generated at the molecular level. The model posits that each gene within a genetic clock or a genetic cascade is regulated by two enhancers: one 'dynamic' that induces rapid changes in gene activity, and another 'static' that stabilizes it. By modulating the balance between the potency of dynamic vs static enhancers, the tuning of the speed of gene regulation is achieved (**Figure 1C and D**). We first characterized the model's predictions for the spatiotemporal activities of enhancer reporters of dynamic and static enhancers (**Figure 5**). The model predicts that reporter constructs of dynamic enhancers would drive a gene expression wave that progressively decreases in intensity in the direction of its propagation, whereas reporter constructs of static enhancers would drive a wave whose intensity increases in the direction of its propagation (**Figure 5A**). We then used our enhancer discovery

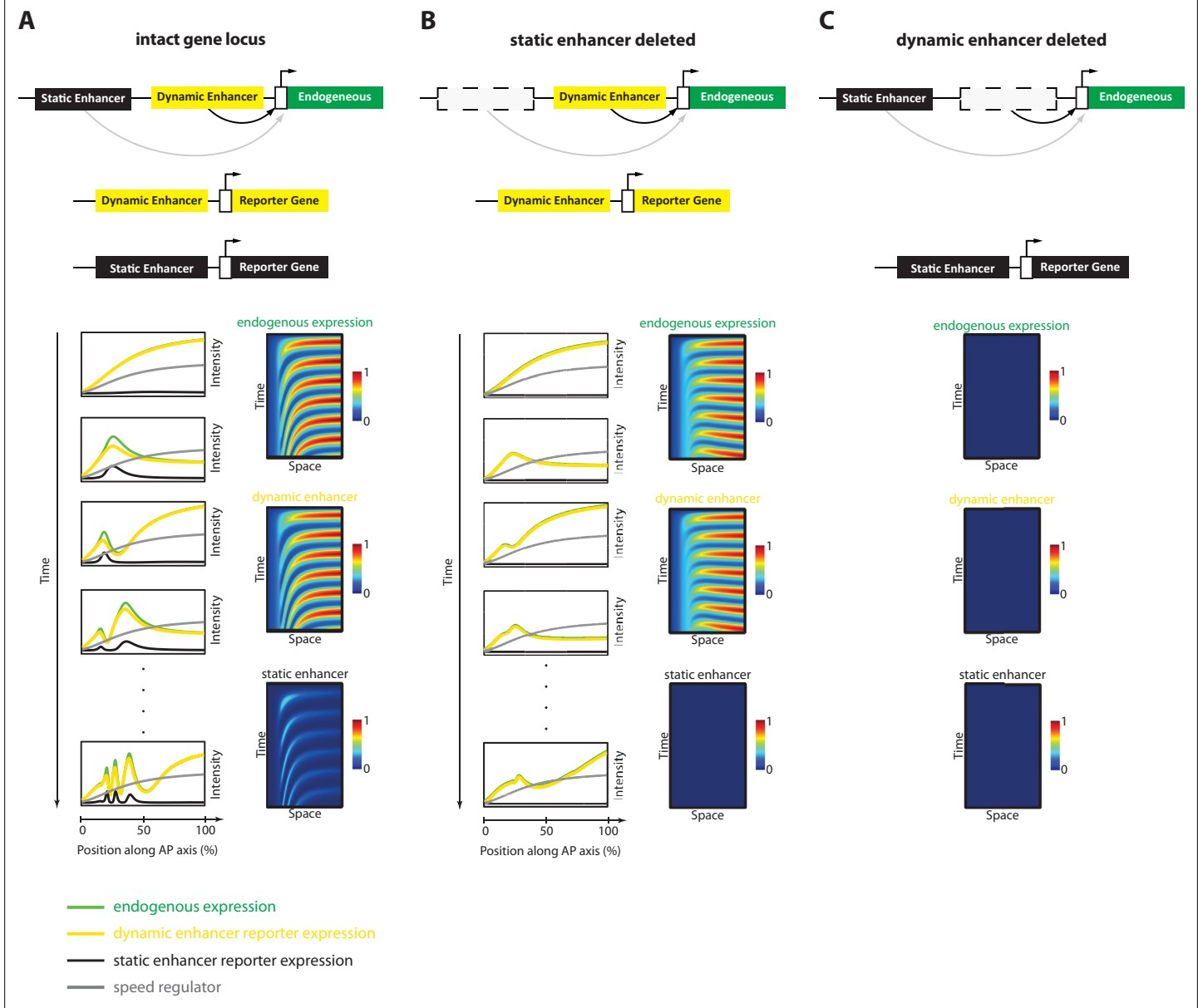

**Figure 7.** Simulation of the Enhancer Switching Model with deleted enhancers. Shown simulation outputs of the Enhancer Switching model under three experimental conditions: (**A**) an intact locus, (**B**) a locus in which the static enhancer is deleted, and (**C**) a locus in which the dynamic enhancer is deleted.

framework to discover a number of enhancers regulating embryonic patterning in the early *Tribolium* embryo (*Figure 4*). One of these enhancers, runB, drove an expression pattern consistent with a role as a static enhancer for the pair-rule gene *run* (*Figure 6A–D*). Another enhancer, hbA, drove an expression pattern consistent with a role as a dynamic enhancer for the gap gene *hb* (*Figure 6E*). However, the expression pattern driven by hbA could be also interpreted as driving the entirety of *hb* expression (*Figure 6E*, *Figure 5B*). We present these findings as tentative support for the Enhancer Switching model, whereas a strong support requires: (*Spitz and Furlong, 2012*) finding enhancer pairs for several gap and pair-rule genes whose activity dynamics match those predicted for static and dynamic enhancers, and (*Shlyueva et al., 2014*) verifying that the deletion of either dynamic or static enhancers result in phenotypes predicted by the model (*Figure 7*). Specifically, deleting a static enhancer should reduce the gene expression wave into an (almost) homogenously oscillating (or sequentially activating) domain at the posterior, that fails to resolve into gene expression bands anteriorly (*Figure 7B*), while deleting a dynamic enhancer should abolish the entire gene expression

(*Figure 7C*). Future works should aim at testing these predictions, modifying the model, or finding alternative models. Furthermore, time- and tissue-specific ATAC-seq analysis should be extended to include more datapoints in time and space in order to make significant correlations between the chromatin landscape and specific developmental events, possibly using single cell ATAC-seq, for which the results of the current study could be used to resolve the position of single cells within the embryo (*Reddington et al., 2020*).

## Materials and methods

### Beetle cultures

Beetle cultures were reared on flour supplemented with 5% dried yeast in a temperature- and humidity-controlled room at 24 °C. To speed up development, beetles were reared at 32 °C.

### PiggyBac enhancer reporter constructs

A piggyBac plasmid with the 3 x P3-mCherry/mOrange marker construct and multiple cloning sites (*Strobl et al., 2018*) was used to generate all enhancer constructs in this study. For enhancer constructs, putative enhancer regions, the *Drosophila* synthetic core promoter (*Lai et al., 2018*; *Pfeiffer et al., 2008*), and the MS2-*yellow* reporter gene (*Garcia et al., 2013*; *Bothma et al., 2014*) were amplified by PCR, assembled through ligation, and inserted into the multiple cloning site of the piggyBac plasmid. Used primers are listed in *Supplementary file 1*.

### Creation of the MCP-mEmerald construct

An artificial sequence, consisting of (i) an AscI restriction enzyme site, (ii) the *Tribolium castaneum* Georgia2 (GA2) background strain-derived (*Richards et al., 2008*) tubulin alpha 1-like protein (aTub) promoter (*Siebert et al., 2008*), (iii) a *Tribolium castaneum* codon-optimized open-reading frame consisting of (a) the SV40-derived nuclear localization signal (NLS) tag (*Chook and Blobel, 2001*) coding sequence, (b) the human influenza hemagglutinin (HA) tag (*Field et al., 1988*) coding sequence, and (c) the bacteriophage MS2 coat protein (MCP) (*Bertrand et al., 1998*) coding sequence, and (iv) a NotI restriction enzyme site, was de novo synthesized and inserted into the PacI/SacI restriction enzyme site pair of pMK-T (Thermo Fisher Scientific) by the manufacturer. The resulting vector was termed pGS[aTub'NLS-HA-MCP]. The sequence was excised with AscI/NotI and inserted into the backbone of the accordingly digested pACOS{#P'#O(LA)-mEmerald} vector (*Strobl et al., 2018*). The resulting vector was termed pAGOC{aTub'NLS-HA-MCP-mEmerald}, contained (i) an expression cassette for mEmerald-labeled (*Shaner et al., 2005*) and NLS/HA-tagged MCP, (ii) the piggyBac 3' and 5' inverted terminal repeats (*Li et al., 2005*), as well as (iii) mOrange-based (*Shaner et al., 2008*) and mCherry-based (*Shaner et al., 2004*) eye-specific (*Berghammer et al., 1999*) transformation markers, and was co-injected with the standard piggyBac helper plasmid (*Handler and Harrell Ii, 1999*) into *Tribolium castaneum* embryos following standard protocols (*Lorenzen et al., 2003*; *Berghammer et al., 2009*) to achieve germline transformation.

### *Tribolium* transgenesis

PiggyBac constructs were transformed into vermilion[white] (*Lorenzen et al., 2002*) with mCherry/mOrange as visible makers. Germline transformation was carried out using the piggyBac transposon system (*Handler and Harrell Ii, 1999*; *Handler, 2002*).

### Egg collections for developmental time windows

Developmental time windows of 3 hr were generated by incubating 3 hr egg collections at 24 °C for the desired length of time. Beetles were reared in flour supplemented with 5% dried yeast.

### In situ hybridization, antibody staining, and imaging of fixed embryos

In situ hybridization was performed using the third-generation in situ hybridization chain reaction (HCR) method (*Choi et al., 2018*). All probe sets and hairpins were ordered at Molecular Instruments. Lot numbers of probe sets are as follows: PRA978 (*run* mRNA), PRA979 (*hb* mRNA), PRC655 (*yellow* mRNA), and PRE723 (*yellow* intron). Antibody staining for Cad protein (primary antibody: Rabbit anti-*Tribolium*-Caudal (*Schulz et al., 1998*), secondary antibody: TRITC - Goat anti-Rabbit-IgG (Jackson

ImmunoResearch Europe Ltd, Ely (UK))) was done following in situ HCR staining. Images were taken with a Leica SP5 II confocal. A magnification of 20 x or 63 x was used at a resolution of 2048 × 1024. Images were processed and enhanced for brightness and contrast using Fiji (*Schindelin et al., 2012*).

## Live imaging

aTub >MCP-mEmerlad females were crossed with hbA >MS2-*yellow* or runB >MS2-*yellow* males. Three hours of egg collections were generated and incubated for eleven (crossing with hbA) or 14 hr (crossing with runB) at 24 °C. Embryos were dechorionated by immersion in 1% bleach for 30 s twice. Embryos were mounted using the hanging drop method and covered with halocarbon oil 700 (Sigma). Time-lapse movies were taken by capturing 41 planes every 3 min over ~6 hr at 21 °C with a Leica SP5 II confocal. A magnification of 63 X was used at a resolution of 1024 × 900. To produce unprocessed live imaging movies (*Video 1* and *Video 4*), a maximum Z-projection is applied to the image sequence in Fiji.

## Computational processing and analysis of live imaging movies

To characterize the transcription dynamics driven by enhancer MS2 enhancer reporters in live embryos, circumventing the ambiguity introduced by nuclear flow, we developed a computational strategy to: (*Spitz and Furlong, 2012*) segment nuclei, (*Shlyueva et al., 2014*) detect MS2 spots and estimate their intensity, (*Small et al., 1992*) associate MS2 spots to nuclei and track nuclei over time, and estimate mRNA intensity.

### Segmenting nuclei

In Fiji, stacks were the first maximum intensity projected. Contrast was enhanced using the CLAHE plugin with a block size of 128. Nuclei were then detected as local maxima, disregarding maxima with an intensity below half the image maximum intensity. Detected maxima were used as seed points for the watershed algorithm to retrieve nuclei outline.

### MS2 spot detection

In Fiji, MS2 spots were detected as local 3D maxima after applying a 3D Difference-of-Gaussians filter. Its parameters, the standard deviations of the Gaussians, tolerance (the minimum intensity difference between neighbor spots, analogous to the ImageJ 'Find Maxima' implementation), and a lower threshold (to disregard spots with low intensity) were set empirically.

### Tracking nuclei and MS2 spots over time and mRNA estimation

We used strategies similar to those described in *El-Sherif and Levine, 2016* using Matlab.

## ATAC-seq library preparation

Embryos of the nGFP line (*Sarrazin et al., 2012*) were collected in flour supplemented with 5% dried yeast for 3 hr and incubated for 23–26 hr AEL or 26–29 hr AEL at 24 °C. Embryos were dechorionated by immersion in 1% bleach for 30 s twice. Selected embryos were dissected into three parts (anterior, middle, and posterior). For each biological replicate, three of the same embryo parts were pooled, and three replicates were prepared for each sample condition. Library preparation was performed as previously described (*Buenrostro et al., 2015*; *Blythe and Wieschaus, 2016*). Tagmentation was performed for 8 min. A total of 18 ATAC-seq libraries (3 regions × 2 time points × 3 replicates) were sequenced on an Illumina NovaSeq 6000 at the Novogene Cambridge Genomic Sequencing Centre. 2 × 150 bp paired-end Illumina reads were obtained for all sequenced ATAC-seq libraries.

## ATAC-seq data pre-processing

Sequencing reads were trimmed with cutadapt (*Martin, 2011*) with parameters '-u 15 U 15 -q 30 m 35 --max-n 0 -e 0.10 a CTGTCTCTTATA -A CTGTCTCTTATA' to remove adapter sequences and mapped to the *Tribolium castaneum* reference genome (Tcas5.2, GCA_000002335.3) with BWA-MEM (version 0.7.12-r1039 *Li, 2013*). Next, read duplicates were marked with Picard MarkDuplicates (version 2.15.0, Picard Toolkit. 2019. Broad Institute, GitHub Repository. https://broadinstitute.github.io/picard/; Broad Institute.). Low-quality and duplicated reads were filtered using samtools

view (version 1.10, *Danecek et al., 2021*) with parameters '-F 1804 f 2 -q 30.' To flag regions that appear to be artifacts, we generated a blacklist using a strategy similar to the one developed by the ENCODE Project (*Amemiya et al., 2019*). Specifically, the genome was first divided into non-overlapping 50 bp bins. Next, the BAM files containing the filtered mapped reads were converted into BigWig files using BAMscale (version 1.0, *Pongor et al., 2020*) with parameters 'scale `--oper-ation` unscaled `--binsize` 20 `--frag.`' Using the resulting BigWig files, the mean signal for each bin was computed across all sequencing libraries. Finally, bins with a mean signal equal to or greater than 100 were flagged as artifacts and included in a 'blacklist.' The threshold was determined by visual inspection of the distribution of the mean signals. Reads mapping to genomic regions in the blacklist were filtered out using samtools view (version 1.7, *Danecek et al., 2021*) with parameters '-L' and '-U.'

Peaks were called on individual replicates using macs3 (v3.0.0a7, *Zhang et al., 2008a*; *Zhang et al., 2008b*) callpeak with parameters '-g 152413475 -q 0.01 f BAMPE.' One library was excluded from further analyses upon quality control. The sets of peaks called in each sample were then compared to each other and merged if they overlapped by at least 1 bp. Only merged peaks supported by peaks called in at least two different samples and on scaffolds assigned to linkage groups were considered for subsequent analyses. These peaks are further referred to as the set of 'all' (consensus) chromatin-accessible sites.

Chromatin-accessible sites were annotated with HOMER (version 4.11.1, *Heinz et al., 2010*) using the annotatePeaks.pl function.

### Genomic tracks

Normalized ATAC-seq coverage tracks were generated with BAMscale (version 1.0, *Pongor et al., 2020*) using parameters 'scale `--frag --binsize` 20 `--smoothen` 2.' The tracks of different biological replicates of the same sample were then merged with the UCSC tools bigWigMerge and bedGraphToBigWig (http://hgdownload.cse.ucsc.edu/admin/exe/linux.x86_64/), and visualized with pyGenomeTracks (version 3.7, *Lopez-Delisle et al., 2021*; *Ramírez et al., 2018*).

### Differential accessibility analysis

Differential accessibility analysis of the sites between different regions of the germband and/or time points was performed using the edgeR (version 3.36.0, *Robinson et al., 2010a*) and DESeq2 (version 1.34.0, *Love et al., 2014*) methods within the DiffBind (version 3.4.11, *Ross-Innes et al., 2012*), https://bioconductor.org/packages/release/bioc/vignettes/DiffBind/inst/doc/DiffBind.pdf. Peaks with a false discovery rate (FDR) less than or equal to 0.05 for edgeR and/or DESeq were considered significant. Read counts for each peak were quantified with the dba.count() function, with parameters 'score = DBA_SCORE_TMM_MINUS_FULL, fragmentSize = 171, bRemoveDuplicates = TRUE.' Briefly, this normalizes the read counts using Trimmed Mean of M-values (TMM, *Robinson and Oshlack, 2010b*) scaled by the full library size. We further refer to these values as 'accessibility scores.' Sites that had a mean accessibility score larger than the median and a standard deviation larger than the 3rd quartile were defined as the 'most variable accessible sites.'

PCA on the accessibility scores was performed using the PCA() function in the 'FactoMineR' R package with default parameters (*Lê et al., 2008*).

### Clustering

The z-score normalized accessibility scores of the differentially accessible sites were clustered using k-means (with k=6) as implemented in the kmeans() R function. The function was executed with default parameters except for 'nstart = 1000,' which initializes the algorithm with 1000 different sets of random centers and settles for those giving the best fit.

Complete linkage on the pairwise Pearson correlation distances (1-Pearson correlation coefficient) was performed with the hclust() R function to hierarchically cluster the differentially accessible sites based on their average accessibility scores across the replicates of each sample group. The pheatmap (*Kolde, 2019*) R package was used for visualization using row-scaling.

### Overlap between differentially accessible sites and constructs

A construct was considered to be overlapping with a site if at least 90% of the site overlapped with the construct.

## Computational modeling

For the development of our Enhancer Switching model presented in this study, we implemented computational algorithms as described in references (*Zhu et al., 2017*; *Boos et al., 2018*; *Kuhlmann and El-Sherif, 2018*). We utilized the Matlab programs associated with these publications, incorporating slight modifications (e.g. to simulate enhancer deletion experiments shown in *Figure 7*). What follows is a succinct description of the computational models presented in this study.

For our simulations, we used the 3-genes GRN architecture in the transcription rate of each gene (*G*) is proportional to the weighted sum of two enhancer activities: the dynamic enhancer (*D*), and the static enhancer (*S*), plus an mRNA decay term ($\lambda$).

$$\frac{dG_1(t)}{dt} = \alpha_1 D_1(t) + \beta_1 S_1(t) - \lambda t$$

$$\frac{dG_2(t)}{dt} = \alpha_2 D_2(t) + \beta_2 S_2(t) - \lambda t$$

$$\frac{dG_3(t)}{dt} = \alpha_3 D_3(t) + \beta_3 S_3(t) - \lambda t$$

where α and β are the strength of activity of the dynamic and static enhancers, respectively.

The dynamic enhancers (*D*) encode the wiring of a clock, whereas static enhancers (*S*) encode the wiring of a multi-stable network. We used specific wiring and parameters for these schemes, but other schemes with a wide variety of parameters work as well (*Zhu et al., 2017*; *Kuhlmann and El-Sherif, 2018*).

$$\frac{dD_1(t)}{dt} = R \frac{1}{1 + G_2^n(t)}$$

$$\frac{dD_2(t)}{dt} = R \frac{1}{1 + G_3^n(t)}$$

$$\frac{dD_3(t)}{dt} = R \frac{1}{1 + G_1^n(t)}$$

$$\frac{dS_1(t)}{dt} = (1 - R) \frac{G_1^n(t)}{1 + G_1^n(t)} \cdot \frac{1}{1 + G_2^n(t)} \cdot \frac{1}{1 + G_3^n(t)}$$

$$\frac{dS_2(t)}{dt} = (1 - R) \frac{1}{1 + G_1^n(t)} \cdot \frac{G_2^n(t)}{1 + G_2^n(t)} \cdot \frac{1}{1 + G_3^n(t)}$$

$$\frac{dS_3(t)}{dt} = (1 - R) \frac{1}{1 + G_1^n(t)} \cdot \frac{1}{1 + G_2^n(t)} \cdot \frac{G_3^n(t)}{1 + G_3^n(t)}$$

Where R is the Speed Regulator and is expressed as a gradient in space. The activation of dynamic enhancers by the speed regulator is represented by the multiplication by *R*, where the repression of static enhancers by the speed regulator is represented by the multiplication of the inverse gradient ($1 - R$), where *R* has a maximum magnitude of 1. Other representations of the activating and repressing effects of R on dynamic and static enhancers work as well (*Zhu et al., 2017*; *Kuhlmann and El-Sherif, 2018*).

in silico reporter gene experiments (*Figures 5 and 7*) were carried out for $G_2$.

$$\frac{dRD_2(t)}{dt} = D_2(t) - \lambda t$$

$$\frac{dRS_2(t)}{dt} = S_2(t) - \lambda t$$

Where $RD_2$ is a reporter gene for the dynamic enhancer of $G_2$, and $RS_2$ is a reporter gene for the static enhancer of $G_2$.

Values of used parameters for simulations in *Figure 5A*: n=5, $\lambda$ =0.5, $\alpha_1 = \alpha_2 = \alpha_3 =$ 1.5, $\beta_1 = \beta_2 = \beta_3$ = 1. For *Figure 5B*: n=5, $\lambda$ =0.5, $\alpha_1 = \alpha_2 = \alpha_3 =$ 1.5, $\beta_1 = \beta_2 = \beta_3$ = 0.5. For *Figure 7B*: n=5, $\lambda$ =0.5, $\alpha_1 = \alpha_2 = \alpha_3 =$ 1.5, $\beta_1 = \beta_3$ = 0.5, $\beta_2$ = 0. For *Figure 7C*: n=5, $\lambda$ =0.5, $\alpha_1 = \alpha_3 =$ 1.5, $\alpha_2 = 0$, $\beta_1 = \beta_2 = \beta_3$ = 0.5.

## Acknowledgements

This work is supported by a DFG grant (EL 870/2–1) to EE, and a doctoral fellowship from the German Academic Scholarship Foundation to CM. We thank Rodrigo Nunes da Fonseca, Emilila Esposito, Alexander Aulehla, and Shelby Blythe for providing valuable tips for ATAC-seq library preparation.

## Additional information

### Funding

| Funder | Grant reference number | Author |
|---|---|---|
| Deutsche Forschungsgemeinschaft | EL 870/2-1 | Ezzat El-Sherif |
| Studienstiftung des Deutschen Volkes | | Christine Mau |

The funders had no role in study design, data collection and interpretation, or the decision to submit the work for publication.

### Author contributions

Christine Mau, Conceptualization, Funding acquisition, Investigation, Writing - original draft; Heike Rudolf, Timo Regensburger, Investigation; Frederic Strobl, Benjamin Schmid, Methodology, Writing - original draft; Ralf Palmisano, Ernst HK Stelzer, Resources; Leila Taher, Conceptualization, Investigation, Writing - original draft; Ezzat El-Sherif, Conceptualization, Supervision, Funding acquisition, Investigation, Writing - original draft

### Author ORCIDs

Benjamin Schmid http://orcid.org/0000-0002-9327-2296
Ralf Palmisano http://orcid.org/0000-0003-4283-2115
Ernst HK Stelzer http://orcid.org/0000-0003-1545-0736
Leila Taher http://orcid.org/0000-0002-2013-5426
Ezzat El-Sherif http://orcid.org/0000-0003-1738-8139

### Decision letter and Author response

Decision letter https://doi.org/10.7554/eLife.84969.sa1
Author response https://doi.org/10.7554/eLife.84969.sa2

## Additional files

### Supplementary files

• Supplementary file 1. List of used primers. frw: forward, rev: reverse.

• MDAR checklist

### Data availability

Raw sequence files and scaled coverage tracks (in bigWig format) have been deposited in the Gene Expression Omnibus database under accession number GSE235410. Scaled coverage tracks were also uploaded to the iBeetleBase Genome Browser (https://ibeetle-base.uni-goettingen.de/genome-browser/) (*Dönitz et al., 2018*). Matlab codes for the Speed Regulation and Enhancer Switching models can be found in *Kuhlmann and El-Sherif, 2018*, and can be modified based on information and parameter values indicated in Materials and Methods (Computational Modeling) to generate the simulations presented in this study. Generated transgenic Tribolium lines are available upon request.

The following dataset was generated:

| Author(s) | Year | Dataset title | Dataset URL | Database and Identifier |
|---|---|---|---|---|
| Mau C, Rudolf H, Strobl F, Schmid B, Regensburger T, Palmisano R, Stelzer E, Taher L, El-Sherif E | 2023 | ATAC-seq analysis of chromatin accessibility in Tribolium castaneum germband stage embryos | https://www.ncbi.nlm.nih.gov/geo/query/acc.cgi?acc=GSE235410 | NCBI Gene Expression Omnibus, GSE235410 |

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
