## [Editor Report]

The authors describe a sophisticated method to follow enhancer activity in both live embryos and fixed embryos in Tribolium and present important data about the function of a number of enhancers in early development. They show that some of the enhancers are "dynamic" and others are "static" and use this to provide solid support for the "enhancer-switching" model of gene regulation suggested by some of these authors in the past. Additional work is required to provide conclusive validation of the "enhancer switching" model.

---

## [Decision Letter]

**Decision letter after peer review:**

Thank you for submitting your article "How enhancers regulate wavelike gene expression patterns: Novel enhancer prediction and live reporter systems identify an enhancer associated with the arrest of pair-rule waves in the short-germ beetle Tribolium" for consideration by *eLife*. Your article has been reviewed by 3 peer reviewers, including Ariel Chipman as a Reviewing Editor and Reviewer #1, and the evaluation has been overseen by Claude Desplan as the Senior Editor.

All three reviewers were impressed by the technical advances presented in the manuscript and agreed that this was its main contribution. However, all three also felt that the framing of the manuscript was not correct and that the authors are making stronger claims than what the data support. While no new data or experiments are needed, a revised manuscript should include more details about the methods and results, as detailed in the reviewers' comments. More importantly, the revised manuscript should stick to the data and be more cautious when making claims about the possible interpretations of the results. The manuscript in its current form reads too much like a review paper and not enough like a paper presenting important new tools and results. Significant rewriting is necessary before the manuscript can be accepted for publication.

Essential revisions:

1) Significantly shorten the Introduction, omitting all superfluous discussion of enhancers, Tribolium tools, and approaches to ATAC-seq, beyond what is necessary to understand the results.

2) provide more data about the activity of the studied enhancers at later stages, or if no such data is available, explain why you limited your study to these stages.

3) Provide better documentation of the experimental procedure, both in figures and in text.

4) Tone down the claims about the support your data provide to the enhancer switching model. The results are consistent with the model but do not actually provide direct support for the model.

5) Correct the errors in the text pointed out by all three reviewers.

6) Modify the figures to make them color-blind-friendly.

*Reviewer #1 (Recommendations for the authors):*

Line 49 – Somitogenesis does not demarcate future vertebrae, but somites, of which vertebrae are only one resulting tissue.

Line 57 – Tribolium retains an ancestral mode, it doesn't adopt it.

Line 63 and elsewhere – there is a recurring confusion between dynamic regulation and dynamic expression. These are very different concepts, but they seem to be used interchangeably.

Various places – you often use "expressions" in the plural. This is incorrect and should be replaced by "expression patterns" or "expression domains".

Line 87 – Why assume two enhancers per gene, when you can assume two roles for one enhancer?

Line 99-101 – The dynamics of *Drosophila* gene expression are assumed to be vestigial. We cannot be sure. What are "canonical" expression waves? Most people mean *Drosophila* when referring to anything canonical, but you seem to mean non-*Drosophila* patterns are canonical. I think both views are actually incorrect.

Lines 194-212 – I had to read through this paragraph several times to understand that "cluster 1,408 sites" is not referring to over fourteen hundred sites, using unclear grammar, but to four hundred sites in cluster 1. There should be a space there.

Lines 200-210 – There is a discussion of the percentage of sites with variable accessibility. They seem to be distributed fairly randomly, suggesting that there is no biological significance. Is there a null model for expected differences in accessibility that would allow a test regarding the relevance of this variability? I suspect there isn't and suggest removing this whole discussion.

Lines 210-212 – I wouldn't attach too much significance to differences between GO terms at this level. The difference between "pattern specification process" and "anterior/posterior pattern specification" is likely to be relevant for *Drosophila*, but there is no reason to assume any relevance in Tribolium.

Line 253 – It's a bit strange that the hb domain which is defined as an anterior domain is found in the posterior of the embryo. Please find a clearer way to define this domain.

Line 263 – Puncta is plural. The singular is punctum.

Line 326 – The model shows a shift of expression towards the anterior. However, at the same time, the embryo is extending posteriorly. In reality, the relative position of the expression domains will be shifting, but their absolute position should not be shifting. Does the model take this into account?

Figures:

Figure 1: Odd choice of colors. The red and green in panel B are probably inaccessible to colorblind readers. The yellow in panels C and D is pretty shocking. Maybe patterns instead of colors?

Figure 2: Panel A – It is good that you actually used photos of the cut embryos, but they should be accompanied by a schematic figure for people who do not know the embryos well enough to understand what is being shown. Are panels C, and D necessary? What do they tell us that a table or text wouldn't? Panel F – see comment in the main text. Is this biologically relevant?

Figure 3: Excellent figure with very cool results. Scale bars and orientation (A-P axis) would make it even better.

Figure 4: Scale bars and orientation as above. Panel A is almost unreadable at this size. Are panels D, and E necessary? They would be better as a table.

Figure 6: Scale bars and orientation. What part of the embryo is Panel C showing? Panels D, and E – some units along the axes would be helpful.

*Reviewer #2 (Recommendations for the authors):*

Detailed comments:

Please change red/green depictions to red/magenta or other colors in order to allow colleagues with a red-green deficiency to judge the panels.

Introduction

I am not sure how happy I am with the first paragraph of the introduction. On several instances I hesitated to fully agree with the wording:

– "…embryo is growing…" the challenge is in the first place pattern formation – not growth, right?

– "…decision making relies on … enhancers…" rather: "…on differential gene expression mediated by …"

– Current view that genes are activated in stable domains with little shifts. I would say that dynamics have been described a lot such that this "current view" is not so current anymore: shift and sharpening of gap gene domains in *Drosophila*, activation of broad domains based on morphogens with subsequent sharpening by cell-autonomous GRNs in the vertebrate neural tube, etc.

– "…sweep the embryo…" – only the posterior growth and elongation zone, right?

– Vertebrate segmentation in the first place defines somites – vertebrae emerge secondarily from them.

– Dynamics are not only introduced by wave-like expression but also by temporal switches in development, e.g. the switch of pair-rule gene networks by opa.

Actually, I suggest re-writing this paragraph completely thinking once more about what the key open question is and what previous knowledge most readers will have. Not sure if everyone knows what you mean by "static" vs. "dynamic" enhancers. Maybe first introduce hierarchical vs. dynamic patterning and then the underlying assumption on enhancer function according to your model (static vs. dynamic)?

A brief reminder of the *Drosophila* hierarchical segmentation paradigm would help before the more recent dynamic view of segmentation is introduced. Many researchers trained on flies would probably wonder when you claim that "in insects" pair-rule genes do the segmentation while gap genes (just?) do the regionalization.

The opa example might be mentioned as a related phenomenon.

Line 68 ff Thanks for reminding the reader of the enhancer switching model. For sake of clarity, I suggest explaining and visualizing only the periodic patterning (because the work mainly deals with that aspect). At the end of that paragraph, you could mention that the model also works for non-periodic patterning with respective references.

line 103-107 Shorten. I see the advantage of RNAi in that species but fewer words would suffice.

L 114 Please add the specific expectation for the two types of enhancers that the model predicts.

L 118 ff: Shorten. These important points can be made much more briefly.

L 132.141 Shorten: I think that no review of available methods is required – just the advantage of the new method. Are the MS2 reporters recruited to the "transcription start site" based on the number of polymerases or rather to "mRNAs being transcribed from that locus"?

L 151-159 I think the biological finding should be more prominently mentioned here – What was found? In how far is this in line with the model? The technical novelties should be mentioned as well but not as the main point.

Results:

Please provide pictures of embryos of the two stages for which you did ATAC-seq – this will be important for others to judge, whether your data is useful for their questions.

Line 174-212 In these two paragraphs, many details are listed one after the other such that take-home messages might be missed. Maybe it would help to structure this part according to the main questions/messages that are formulated before showing the data (e.g. more AP differences than timing; anterior region distinguished from middle and post region; dynamics etc.). Please reorganize such that details that do not contribute to one of those main messages are documented in the supplementary.

Line 227-239/249-250 Too many technical details – why not just a reference to the figure/figure legend and focus on the main points (similar to the paragraph 240-248).

Line 298/299 Is this differential accessibility criterion new? Then state more explicitly. If it is based on previous findings in other organisms then cite the respective work.

319-330 is this new or has this been shown in the previous paper already? Or: what aspect is new?

Discussion:

How does your enhancer prediction based on enhancer dynamics relate to previous efforts by Lai et al.?

433 ff The remainder of the discussion contains very thoughtful insights into expression waves, the enhancer switching model, and alternatives. However, it has remained unclear to me what your work actually adds to these issues. This discussion seems a bit unconnected with the actual work as it appears that nothing is being added besides the establishment of Tribolium as a model system for these questions. One condensed brief paragraph showing the potential of your beetle system might suffice for that purpose. Some of the thoughts could be used in the introduction for explaining the motivation for establishing the system.

Figures:

MS2 pictures (Figure 3 and suppl. Figure 9): I had a hard time seeing the puncta in my printout. Might help to adjust the brightness contrast to some degree.

Figure 4C is too small to judge well. Could extend across the whole width while D and E can be made smaller. A closeup of the neural expression might be helpful to judge better.

Suppl. Figures seem not to be cited in order.

Suppl. Figure 10: I did not see the stripes of endogenous runt and yellow intronic probe well enough to judge.

*Reviewer #3 (Recommendations for the authors):*

There is a remarkably low extent of change in accessibility between time points, in the same spatial region. Please comment on whether such a high level of temporal consistency has been reported in previous reports of accessibility during embryonic development (e.g., in whole embryo studies).

Nine enhancers were tested using reporters, of which four drove expression. This includes one previously reported enhancer. Thus, it seems three new enhancers were validated functionally. Is this correct?

The statistics of Figure 4E are on small-sized samples, so the association needs statistical evaluation. I did such a test (Fisher's exact) with the numbers (5,1,3,2) and found the p-value to be ~0.5, clearly not significant.

In the last subsection of results, the concept of "static" and "dynamic" enhancer needs to be better explained to the reader who may not be familiar with prior work. In particular, what is not clear is how the circuits of Figure 1C are simulated at the "enhancer-level". In particular, what is not clear is: in the simulations of Figure 5, which of the three genes in the circuits of Figure 1C are shown (presumably, the red gene), and what is assumed about the dynamics and control of the other two genes? What is meant by weak or strong "static enhancer activity" – some parameter in the simulations? To what extent are the phenomena illustrated in Figure 5 dependent on the parameters assumed for the simulations?

Figure 6 shows interesting and mostly believable evidence in support of one of the predictions of the Enhancer Switching model. However, it was surprising to note that the identity of the "speed regulator" (morphogen) is not yet confirmed and Cad is only "suggested" to play this role. Since the point made by simulations of Figure 5 is tied to the assumed spatial variation of the speed regulator, this needs to be confirmed in order to extract the support the authors need from the results of Figure 6.

---

## [Author Response]

Essential revisions:1) Significantly shorten the Introduction, omitting all superfluous discussion of enhancers, Tribolium tools, and approaches to ATAC-seq, beyond what is necessary to understand the results.

Done.

2) provide more data about the activity of the studied enhancers at later stages, or if no such data is available, explain why you limited your study to these stages.

The dynamics in germband were shown by visualizing yellow mRNA and intronic probes. MS2 imaging was not possible to be used because the embryo dive into the yolk for a while, and then it becomes difficult to capture the germband in the right orientation for imaging. Future work includes using light sheet microscopy for imaging germband stages.

3) Provide better documentation of the experimental procedure, both in figures and in text.

We improved our documentation, especially for computational modeling.

4) Tone down the claims about the support your data provide to the enhancer switching model. The results are consistent with the model but do not actually provide direct support for the model.

See our detailed response to reviewers regarding this issue.

5) Correct the errors in the text pointed out by all three reviewers.

Done.

6) Modify the figures to make them color-blind-friendly.

Done.

Reviewer #2 (Recommendations for the authors):Detailed comments:Please change red/green depictions to red/magenta or other colors in order to allow colleagues with a red-green deficiency to judge the panels.

Done

Introduction– Current view that genes are activated in stable domains with little shifts. I would say that dynamics have been described a lot such that this "current view" is not so current anymore: shift and sharpening of gap gene domains in *Drosophila*, activation of broad domains based on morphogens with subsequent sharpening by cell-autonomous GRNs in the vertebrate neural tube, etc.

See our response to Reviewer #1 regarding this point.

– "…sweep the embryo…" – only the posterior growth and elongation zone, right?– Vertebrate segmentation in the first place defines somites – vertebrae emerge secondarily from them.

Corrected.

– Dynamics are not only introduced by wave-like expression but also by temporal switches in development, e.g. the switch of pair-rule gene networks by opa.Actually, I suggest re-writing this paragraph completely thinking once more about what the key open question is and what previous knowledge most readers will have. Not sure if everyone knows what you mean by "static" vs. "dynamic" enhancers. Maybe first introduce hierarchical vs. dynamic patterning and then the underlying assumption on enhancer function according to your model (static vs. dynamic)?A brief reminder of the *Drosophila* hierarchical segmentation paradigm would help before the more recent dynamic view of segmentation is introduced. Many researchers trained on flies would probably wonder when you claim that "in insects" pair-rule genes do the segmentation while gap genes (just?) do the regionalization.The opa example might be mentioned as a related phenomenon.Line 68 ff Thanks for reminding the reader of the enhancer switching model. For sake of clarity, I suggest explaining and visualizing only the periodic patterning (because the work mainly deals with that aspect). At the end of that paragraph, you could mention that the model also works for non-periodic patterning with respective references.line 103-107 Shorten. I see the advantage of RNAi in that species but fewer words would suffice.

Done.

L 118 ff: Shorten. These important points can be made much more briefly.

Done.

L 132.141 Shorten: I think that no review of available methods is required – just the advantage of the new method.

Done.

L 151-159 I think the biological finding should be more prominently mentioned here – What was found? In how far is this in line with the model? The technical novelties should be mentioned as well but not as the main point.

Done.

Results:Please provide pictures of embryos of the two stages for which you did ATAC-seq – this will be important for others to judge, whether your data is useful for their questions.

Done.

Line 174-212 In these two paragraphs, many details are listed one after the other such that take-home messages might be missed. Maybe it would help to structure this part according to the main questions/messages that are formulated before showing the data (e.g. more AP differences than timing; anterior region distinguished from middle and post region; dynamics etc.). Please reorganize such that details that do not contribute to one of those main messages are documented in the supplementary.

We have substantially shortened the section while trying to preserve the information expected by most readers from a standard ATAC-seq data analysis.

Line 227-239/249-250 Too many technical details – why not just a reference to the figure/figure legend and focus on the main points (similar to the paragraph 240-248).

Done.

Line 298/299 Is this differential accessibility criterion new? Then state more explicitly. If it is based on previous findings in other organisms then cite the respective work.

These lines reflect the interpretation of our own results. In the particular context of our study, differential accessibility is associated with enhancer activity. However, this is unlikely to be a general criterion, since it clearly depends on the time and space scales of interest. Furthermore, while some constitutively accessible sites might represent promoters that are not annotated, others are almost certainly enhancers that either were active in the past, will be active in the future, or would be active under specific conditions. Similar observations have been made for other species. Please, see our response to Reviewer #1’s comments for more details.

319-330 is this new or has this been shown in the previous paper already? Or: what aspect is new?

This is a new simulation. Only a non-periodic version of Figure 5A (but not 5B) was discussed in Zhu et al. PNAS 2017.

Discussion:How does your enhancer prediction based on enhancer dynamics relate to previous efforts by Lai et al.?433 ff The remainder of the discussion contains very thoughtful insights into expression waves, the enhancer switching model, and alternatives. However, it has remained unclear to me what your work actually adds to these issues. This discussion seems a bit unconnected with the actual work as it appears that nothing is being added besides the establishment of Tribolium as a model system for these questions.

These sections are now removed from the revised submission.

Suppl. Figure 10: I did not see the stripes of endogenous runt and yellow intronic probe well enough to judge.

Difficult to make it clear in a print out even by brightening the image. We recommend seeing it on a PC using the Zoom feature in a PDF reader.

Figure 4C is too small to judge well. Could extend across the whole width while D and E can be made smaller. A closeup of the neural expression might be helpful to judge better.

Done

Reviewer #3 (Recommendations for the authors):There is a remarkably low extent of change in accessibility between time points, in the same spatial region. Please comment on whether such a high level of temporal consistency has been reported in previous reports of accessibility during embryonic development (e.g., in whole embryo studies).

We are not aware of any ATAC-seq studies in *Tribolium*, I could not find ATAC-seq studies. A previous FAIRE-seq study (https://doi.org/10.1242/dev.160663) involving some of the authors analyzed three different developmental stages (0-24, 24-48 and 48-72 h) in whole embryos, in the second (T2) and third (T3) thoracic epidermal tissues of the last instar larvae that contain the forewing (elytron) and hindwing imaginal tissues, and in the brain isolated from the last instar larvae. This study found temporal differences in chromatin accessibility, but in general, differences between tissues were orders of magnitude larger than between developmental stages. Of course, a direct comparison is problematic, but these findings are generally in agreement with ours.

Several studies have looked at chromatin accessibility in *Drosophila*. For example, Blythe and Wieschaus (https://doi.org/10.7554/eLife.20148) performed ATAC-seq to measure changes in chromatin accessibility from NC11 to NC13 in 3-minute time intervals. They found that regions of accessibility are established sequentially and that accessibility is stably maintained in highly condensed mitotic chromatin to ensure faithful inheritance of prior accessibility status across cell divisions. Overall, within the temporal interval under consideration, temporal changes in accessibility appeared to be low. Also, Ma and Buttitta (https://doi.org/10.1186/s13072-017-0159-8) used FAIRE-seq to quantify accessibility of regulatory elements in cycling and postmitotic *Drosophila* wings and found virtually no difference. Finally, Thomas et al. (https://doi.org/10.1186/gb-2011-12-5-r43) applied DnaseI to pooled *Drosophila* embryos at corresponding to the transition from the cellular blastoderm (stage 5) through the formation of organ primordia (stages 9, 10, and 11) and the beginning of head involution (stage 14). They observed temporal changes, but most (~55%) DnaseI hypersensitive sites (DHS) sites detected at stage 5 were carried forward through stage 14.

As explained in our response to Reviewer #2, what “dynamic” means strongly depends on the experimental design. If we had compared germband to blastoderm stages we would have most likely identified relatively large changes in accessibility.

Nine enhancers were tested using reporters, of which four drove expression. This includes one previously reported enhancer. Thus, it seems three new enhancers were validated functionally. Is this correct?

Yes.

The statistics of Figure 4E are on small-sized samples, so the association needs statistical evaluation. I did such a test (Fisher's exact) with the numbers (5,1,3,2) and found the p-value to be ~0.5, clearly not significant.

Thank you for your comment. Statistical significance depends on sample size. A small difference may have a statistically significant difference because of a large sample size. Conversely, a large difference in variance may not produce a statistically significant difference if the sample size is small. As the reviewer has pointed out our sample size is small. Unfortunately, as the reviewer is certainly aware of, sufficiently increasing the sample size would involve additional years of experimental work, as well as thousands of Euros. We do not have such resources, but the magnitude of the difference we observed clearly supports our hypothesis that differential expression is associated with enhancer activity. This is just one piece of evidence (e.g., Figure 2E and the functional analysis of the clusters – which is no longer included in the present version of the manuscript – indicate the same). More importantly, similar observations have been done by other colleagues in other species (see our responses to Reviewer #1 and Reviewer #2’s comments). We have rephrased our conclusions, pointing out the limitations of our study.

In the last subsection of results, the concept of "static" and "dynamic" enhancer needs to be better explained to the reader who may not be familiar with prior work. In particular, what is not clear is how the circuits of Figure 1C are simulated at the "enhancer-level". In particular, what is not clear is: in the simulations of Figure 5, which of the three genes in the circuits of Figure 1C are shown (presumably, the red gene), and what is assumed about the dynamics and control of the other two genes? What is meant by weak or strong "static enhancer activity" – some parameter in the simulations? To what extent are the phenomena illustrated in Figure 5 dependent on the parameters assumed for the simulations?

The best way to explain this is through describing the details of the modeling, which we added to the Methods section in the revised manuscript.

Figure 6 shows interesting and mostly believable evidence in support of one of the predictions of the Enhancer Switching model. However, it was surprising to note that the identity of the "speed regulator" (morphogen) is not yet confirmed and Cad is only "suggested" to play this role. Since the point made by simulations of Figure 5 is tied to the assumed spatial variation of the speed regulator, this needs to be confirmed in order to extract the support the authors need from the results of Figure 6.

Previous work (El-Sherif et al. PLOS Genetics 2013 and Zhu et al. PNAS 2017) presented an extensive dataset showing the correlation between Cad gradient and pair rule and gap dynamics, strongly suggesting that Cad/Wnt plays the role of a speed regulation. What is not confirmed is the actual physical binding of Cad to the enhancers of gap and pair-rule genes. We believe however that the available data on the influence of Cad/Wnt gradient is strong enough to assume their role in modulating the timing of gap and pair rule genes in Tribolium.